# Spectrin-beta 2 facilitates the selective accumulation of GABA$_A$ receptors at somatodendritic synapses

Joshua L. Smalley[1], Noell Cho[1], Shu Fun Josephine Ng[1], Catherine Choi [1], Abigail H. S. Lemons[1], Saad Chaudry[1], Christopher E. Bope[1], Jake S. Dengler[1], Chuansheng Zhang[2], Matthew N. Rasband [2], Paul A. Davies[1] & Stephen J. Moss[1,3 ✉]

Fast synaptic inhibition is dependent on targeting specific GABA$_A$R subtypes to dendritic and axon initial segment (AIS) synapses. Synaptic GABA$_A$Rs are typically assembled from α1-3, β and γ subunits. Here, we isolate distinct GABA$_A$Rs from the brain and interrogate their composition using quantitative proteomics. We show that α2-containing receptors co-assemble with α1 subunits, whereas α1 receptors can form GABA$_A$Rs with α1 as the sole α subunit. We demonstrate that α1 and α2 subunit-containing receptors co-purify with distinct spectrin isoforms; cytoskeletal proteins that link transmembrane proteins to the cytoskeleton. β2-spectrin was preferentially associated with α1-containing GABA$_A$Rs at dendritic synapses, while β4-spectrin was associated with α2-containing GABA$_A$Rs at AIS synapses. Ablating β2-spectrin expression reduced dendritic and AIS synapses containing α1 but increased the number of synapses containing α2, which altered phasic inhibition. Thus, we demonstrate a role for spectrins in the synapse-specific targeting of GABA$_A$Rs, determining the efficacy of fast neuronal inhibition.

[1] Department of Neuroscience, Tufts University, Boston, MA 02111, USA. [2] Department of Neuroscience, Baylor College of Medicine, Houston, TX 76706, USA. [3] Department of Neuroscience, Physiology and Pharmacology, University College, London WC1E 6BT, UK. ✉email: Stephen.Moss@tufts.edu

Type A γ-aminobutyric acid receptors (GABA$_A$R) are Cl$^-$ preferring ligand-gated ion channels that mediate fast synaptic inhibition, a process that is critical in determining neuronal output, and limiting hyperexcitability. They are also the sites of action for barbiturates, benzodiazepines and neurosteroids, all of which act as GABA$_A$R-positive allosteric modulators (PAMs).

GABA$_A$Rs are hetero-pentamers that can be constructed from α(1-6), β(1-3), γ(1-3), δ, ε, θ and π subunits[1]. Studies based on knock-out mice and recombinant expression suggest that the majority of synaptic GABA$_A$Rs are composed of α1-3, β1-3 and γ2 subunits[2,3]. Immunological- and benzodiazepine-affinity purification coupled with mass spectroscopy has been used to compare the expression levels of GABA$_A$R α subunit variants in the forebrain. These approaches have demonstrated that α1 is expressed at 5–10-fold higher levels than either the α2 or α3 subunits[4]. Thus, the α1 subunit is likely to be a component of the majority of GABA$_A$Rs that mediate synaptic inhibition in the brain.

Immunolocalization studies have revealed that principal neurons within the hippocampus express α1-3, β1-3 and γ2 subunits, suggesting extensive heterogeneity of GABA$_A$R structure within individual neurons. Consistent with this, receptors containing the α1 subunit are predominantly found at dendritic synapses, while the α2 subunit is highly enriched at their equivalents on the axon initial segment (AIS)[5,6]. Such differential receptor distribution is likely to have profound local effects on excitability and pharmacology as α-subunit isoforms determine the decay of inhibitory postsynaptic currents and ligand sensitivity.

The accumulation of GABA$_A$Rs at synaptic sites is dependent upon their selective confinement at these structures, processes that are dependent upon their direct binding to components of the inhibitory postsynaptic scaffold such as gephyrin (Gphn) and collybistin (CB)[7]. Ablating the expression of either protein or modifying their affinity for individual GABA$_A$R subunits compromises global inhibitory synapse formation and the magnitude of inhibitory synaptic currents[8–10]. But little is known about how the targeting of specific GABA$_A$R subtypes to different neuronal locations is achieved.

To gain insights into the mechanisms that may contribute to the selective accumulation of GABA$_A$Rs at synapses in different neuronal compartments, we isolated biochemically distinct populations of receptors enriched in the α1 and α2 subunits which were then subject to native gel electrophoresis. Their subunit composition and associated proteins were identified and quantified using liquid chromatography coupled with tandem mass spectroscopy (LC-MS/MS) and their composition was subsequently compared.

Here, using this approach, we demonstrate that GABA$_A$Rs assembled from α1 and α2 subunits are likely to comprise of α1α1β3γ2 or α2α1β3γ2, respectively. These receptors are associated with and bind to distinct spectrins; neuronal cytoskeletal proteins, that determine the plasma membrane distribution of integral membrane proteins. Spectrin heterodimers incorporating β2 spectrin (encoded by the *SPTBN1* gene) favored α1 subunit association, whereas β4 spectrin (encoded by the *SPTBN4* gene) specifically associated with the α2 subunit. Ablating β2 spectrin expression altered the β2 spectrin/β4 spectrin balance and compromised the formation of dendritic synapses, but not their counterparts on the AIS, leading to modifications in the magnitude of phasic inhibition.

## Results

### Analyzing the composition of GABA$_A$R subtypes assembled from the α1 and α2 subunits. To gain insights into the composition of endogenous GABA$_A$Rs containing the α1 and α2 subunits, we purified forebrain plasma membranes and

solubilized them in 0.5% Triton 100. We used these extracts to immunoprecipitate α1 and α2 subunit-containing receptors using immobilized α1 monoclonal antibodies and wild-type tissue, or immobilized 9E10 (myc) antibodies and tissue from knock-in mice where the α2 subunit had been modified to contain a pHluorin and 9E10 (myc) epitope on its n-terminus (pHα2) respectively. This modification is functionally silent and required due to the low abundance of the α2 subunit and the paucity of suitable antibodies[11]. The immunoprecipitated proteins were eluted in 2% TWEEN and 0.01% SDS, a combination sufficient to elute the material from the beads without disturbing stable protein complexes (Supplementary Figure 1)[12], resolved by blue native-polyacrylamide gel electrophoresis (BN-PAGE) followed by either colloidal Coomassie staining or immunoblotting with a monoclonal antibody against the α1 subunit (for α1 receptors) or GFP (for α2 receptors). Under these conditions a band of approximately 250 kDa, the predicted molecular mass of pentameric GABA$_A$Rs, was evident together with a larger species of 720 kDa for α2-containing receptors. These species were present in plasma membrane lysates, and following immunopurification, where they were visible by both immunoblot and Coomassie staining (Fig. 1a).

Given that 250 kDa is the predicted molecular mass of a pentameric GABA$_A$R, the composition of this species was analyzed using liquid chromatography coupled with tandem mass spectroscopy (LC-MS/MS). Following subtraction of proteins that bound to anti-9E10 beads in wild type tissue, proteins were quantified via their globally normalized spectral index (S$_I$G$_I$), an accepted means of label-free quantification[13,14]. The 250 kDa band was highly enriched in GABA$_A$R subunits. With purified α2-containing receptors we detected the α2 subunit ($p = 0.009966789$), the α1 subunit was also significantly ($p = 0.017830406$) detected at a 1:1 ratio, along β2 ($p = 0.040276721$), β3 ($p = 0.004662486$), and γ2 ($p = 0.016340521$) (Fig. 1b). To confirm these data were not contamination from closely associated α1-containing receptors, we performed a double immunoprecipitation of α2-containing receptors. Immunoprecipitation was carried out using anti-9E10 beads and pHα2 tissue, followed by elution of the immunoprecipitated proteins, 10-fold dilution with lysis buffer, and a second immunoprecipitation reaction with anti-9E10 beads prior to resolving the complexes by BN-PAGE. The double immunoprecipitated α2-containing protein complexes were clearly visible by Coomassie staining (Fig. 1c). The 250 kDa band was excised and the GABA$_A$R subunit composition was assessed by quantitative proteomics. Strikingly, the α1 subunit was once again significantly detected ($p = 0.000241543$) along with α2 subunits ($p = 0.0007118$), as well as β3 ($p = 0.018429937$), and γ2 ($p = 0.018648054$) subunits (Fig. 1d).

Next, we repeated the experiments described above, but immunoprecipitated α1-containing receptors from wild type tissue using immobilized α1 monoclonal antibodies. α1-containing protein complexes at approximately 250 kDa and 720 kDa and were clearly visible by immunoblot in the total plasma membrane lysates and following immunoprecipitation by both immunoblot and Coomassie staining (Fig. 1e). The 250 kDa species was excised and analyzed by quantitative proteomics. The α1 subunit was recovered ($p = 0.001688864$) at 10-fold higher levels than α2 or α3 subunits, however they were both significantly detected ($p = 0.000595758$ and $p = 0.0162021$ respectively), along with β1 ($p = 0.008332315$), β2 ($p = 0.015409547$), β3 ($p = 0.01113512$), δ ($p = 0.012970599$). and γ2 ($p = 0.008575418$) (Fig. 1f). Components of extrasynaptic GABA$_A$Rs (α4-6) were not detected in either α1 or α2 purifications. Intriguingly, the δ subunit co-purified with α1 (Fig. 1f), which is consistent with published studies suggesting

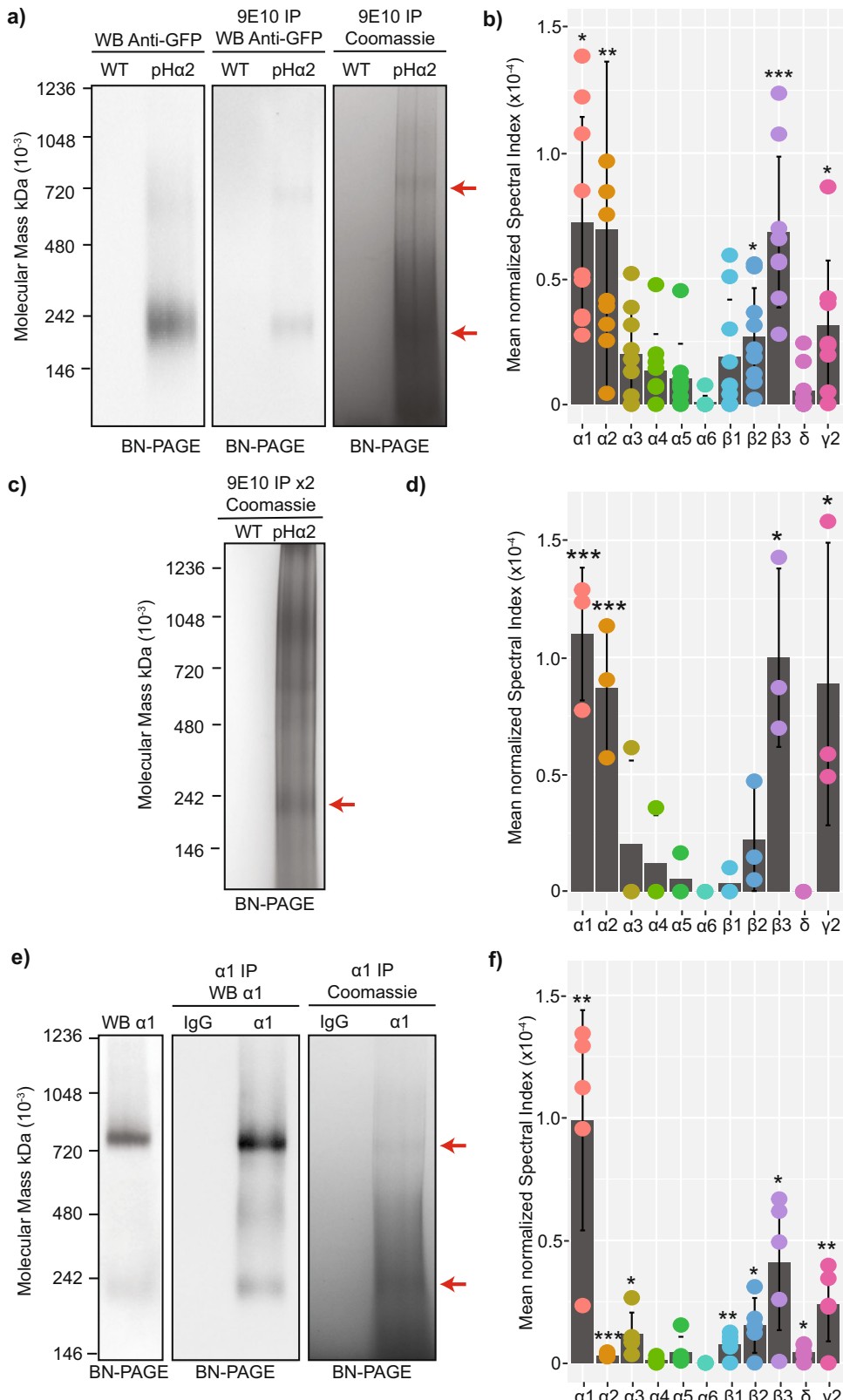

that receptors assembled from α1 and δ subunits contribute to tonic current in some neuronal populations[15].

Collectively, these studies suggest that the majority of synaptic GABA$_A$R subtypes contain the α1 subunit. They further suggest that neurons assemble a highly abundant population of GABA$_A$Rs that contain both α1 and α2 subunits.

**Comparing the subcellular distribution of the GABA$_A$R α1 and α2 subunits.** To interpret our biochemical studies further, we prepared cultured hippocampal neurons from P1 pHα2 mice. The cultures were fixed, permeabilized, and immunostained at DIV18-21 with antibodies against GFP (to enhance the α2 signal), the α1 subunit and Ankyrin G (AnkG); an accepted marker for

**Fig. 1 Analyzing the subunit composition of native GABA_AR subtypes using affinity purification and quantitative mass spectroscopy. a** α2 GABA_AR-containing protein complexes were immunopurified using pre-optimized conditions (Supplementary Figure 1) using anti-9E10 (myc) antibodies, from plasma membrane fractions of pHα2 mice and resolved by BN-PAGE. Stable protein complexes were observed at ~720 kDa and ~250 kDa (red arrows). The complexes were consistently present in the lysate prior to immunopurification. Immunopurified complexes were visualized by both immunoblot and colloidal Coomassie staining. Representative images from an $n = 9$ experiment. **b** The resolved α2 complexes at approximately 250 kDa, the mass of an intact pentameric GABA_AR, were characterized by quantitative proteomics compared to anti-9E10 immunopurified material from wild type mice ($n = 9$, *$p < 0.05$, **$p < 0.01$, ***$p < 0.001$). Detected peptides were mapped to the mouse proteome and the GABA_AR subunit expression quantified by measuring the spectral index normalized to the global index ($S_I G_I$) ($n = 4$). The raw data are contained in Supplementary Data 1. **c** Double immunopurifications for α2-containing GABA_ARs were carried out to achieve more highly purified α2-containing GABA_ARs. These were visualized by immunoblot where a persistent protein band was observed at 250 kDa (red arrow). Representative image from an $n = 3$ experiment. **d** The 250 kDa band was analyzed for GABA_AR subunit expression measured by quantitative LC-MS/MS ($n = 3$, *$p < 0.05$). The raw data are contained in Supplementary Data 2. **e** The immunopurification experiments were repeated for α1 GABA_AR-containing protein complexes in wild type mice. Complexes were resolved and visualized by immunoblot and Coomassie staining. Stable protein complexes were observed at ~720 kDa and ~250 kDa (red arrows). Representative images from an $n = 5$ experiment. **f** The 250 kDa band was analyzed for GABA_AR subunit composition measured by quantitative LC-MS/MS ($n = 5$). Error bars represent the standard error of the mean (SEM). The raw data are contained in Supplementary Data 3. The raw gel/membrane scans are shown in Supplementary Figure 2.

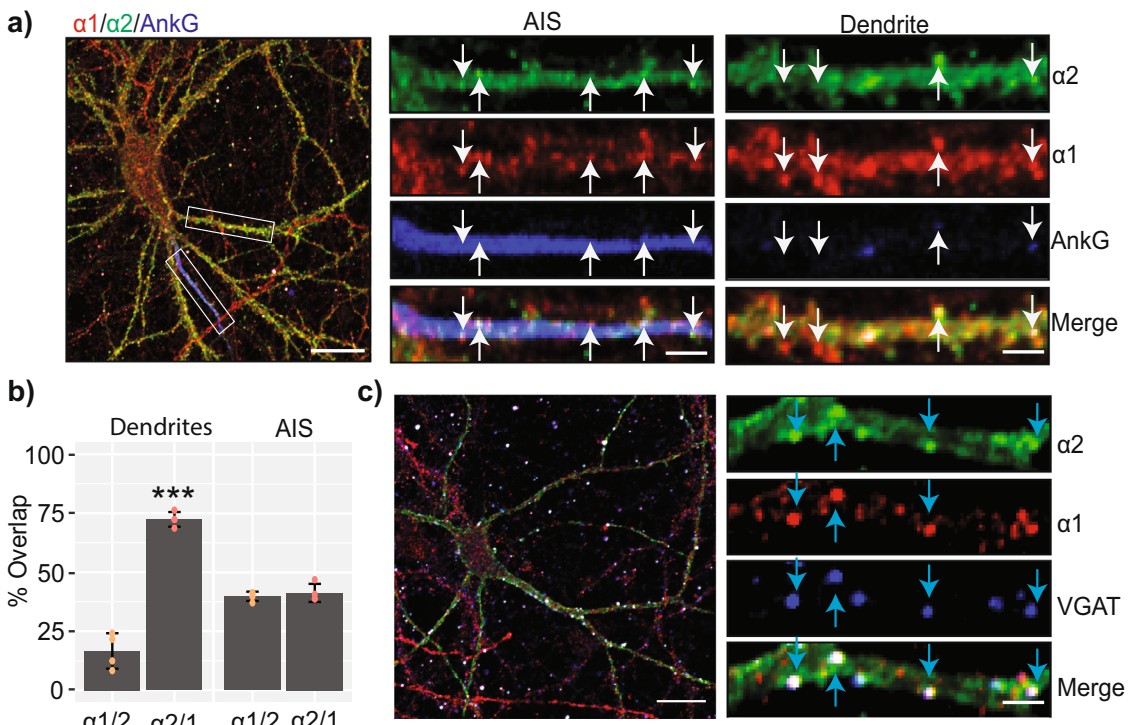

**Fig. 2 Determining the subcellular distribution of α1 and α2 GABA_AR subunits in cultured neurons.** a) Mouse neurons from pHα2 mice were fixed and permeabilized at Days In Vitro (DIV) 21 and immunostained using antibodies against GFP (α2), α1, and AnkG. Dendrites and AIS were distinguished by the presence or absence of AnkG immunoreactivity. Scale bar = 10 μm and 2 μm in cropped images. b) The colocalization of α1 and α2 puncta were quantified in AnkG positive (AIS) and negative (dendritic) compartments. The quantification was expressed as the percentage of α1 puncta that contain α2 (α1/2) and the percentage of α2 puncta that contain α1 (α2/1) ($n = 4$ individual cultures, ***$p < 0.001$). The raw data are contained in Supplementary Data 4. c) DIV21 mouse neurons were immunostained with antibodies against α1, α2 and VGAT to visualize subunit colocalization at active synapses. Scale bar = 10 μm and 2 μm in cropped images. Error bars represent the SEM.

the AIS (Fig. 2a). We compared the co-localization of α1 with α2 subunit immunoreactivity (α1/α2) and vice versa (α2/α1) on the AIS and on cell bodies/dendrites using confocal microscopy. Co-localization of subunit immunoreactivity was then determined using the synapse plugin for ImageJ as previously described (Nakamura et al., 2016, Nathanson et al., 2019 Kontou et al., 2021)[16]. On the AIS approximately 30% of α2 puncta on the AIS immunostaining overlapped with α1 immunoreactivity. Likewise, a similar proportion of α1 puncta containing α2 immunostaining were seen (Fig. 2b). Similar analyses were performed on dendrites which revealed that more than 45% of α2 puncta contained α1 immunoreactivity. In contrast, less than 20% of dendritic α1-

positive puncta contained α2 ($p = 0.0000093$). To ascertain if the respective α subunit isoforms co-localized at synapses, pHα2 cultures were stained with antibodies against the vesicular inhibitory amino acid transporter (VGAT), GFP, and the α1 subunit. Puncta containing both α1 and α2 subunits opposed to VGAT were evident, as were synapses containing α1 subunit-only isoforms (Fig. 2c).

Collectively these results demonstrate high co-localization of the α2 and α1 subunits on both the AIS and dendrites which is consistent with our biochemical studies suggesting their co-assembly into single receptors. The lower levels of co-localization of α1 with α2 in dendrites suggests the existence of a large

population of GABA$_A$Rs containing just the α1 subunit, and a smaller population of mixed α1-α2 receptors in these structures.

**Analyzing the protein composition of high molecular weight α1 and α2 subunit-containing complexes.** We used LC-MS/MS coupled with label-free quantification to compare proteins that co-purify with α1 with α2 containing GABA$_A$Rs. Our studies focused on the 720 kDa species detected using immunoblotting and Coomassie staining following BN-PAGE (Fig. 1a, e). S$_i$G$_i$ values for immunoprecipitated proteins were compared to control purifications (non-immune IgG or anti-9E10 in wild type tissue for α1 with α2 respectively). Only those proteins significantly enriched compared to control were included for downstream analysis. Using these criteria 121 proteins were detected co-purifying with the α1 subunit, while 123 were associated with α2, 45 of which were common to both purifications (Fig. 3a and Supplementary Data 5).

To interrogate the protein complexes further, we performed network analysis on the significantly enriched proteins associated with α1 and α2 receptors. Protein association data was obtained from STRINGdb for known experimental interactions as previously described[14,17,18]. We overlaid the highest-scoring Gene Ontology (GO) Biological Process term for each protein to provide insight into protein function. We also scaled the node size representing each protein to the S$_i$G$_i$ values detected for each protein. In this way we were able to visualize quantitative networks and subnetworks along with developing insights into the functional groups of proteins that associate with GABA$_A$Rs. The network analysis was carried out for the proteins that were exclusively associated with either α1 or α2 receptors, and those that were associated with both receptor subtypes (Fig. 3b). The α1 receptor uniquely co-purified with a significant subnetwork of structural and signaling proteins, including Map6, Actn1, Synpo, Shank3, Ina, Ppp1r9b, Syngap1, and Camk2a. The α2 receptor specifically co-purified with structural proteins, including Cnp, Cntn1, Golga2, Cntnap1, and β4 spectrin. Both α1 and α2 receptors co-purified with a subnetwork of spectrin isoforms; β2 spectrin, β3 spectrin, α2 spectrin, along with Myo5a, Myh10, and Gphn.

Finally, we used principal component analysis (PCA) to assess the degree of similarity in binding protein patterns between the α1 and α2 receptor subtypes. The significantly enriched proteins were assembled into a matrix of all repeats and detected proteins along with the S$_i$G$_i$ values for each. The data were normalized by z-transformation and the PCA plot created using the ggfortify package (accessed January 2021) in R[19]. The α1 and α2 receptors readily divided into two distinct groups based on their associated protein profile. The PCA loadings were also included to illustrate the proteins that were most influential in producing this separation. These included Gphn and β2 spectrin, which were more highly associated with α1 receptors, and β4 spectrin, which was uniquely associated with α2 receptors. Significantly, Ankyrin B and Ankyrin G are known spectrin-binding partners and were also detected in our receptor purifications.

Taken together, these data demonstrate the similarities and the differences in the binding protein profiles of α1 and α2 receptors. They also highlight those specific interactions with different spectrin isoforms may play an important role in determining differential localization of receptor clustering.

**Spectrin isoforms bind differentially to GABA$_A$R alpha subunit isoforms and are enriched at inhibitory synapses.** To investigate the association of α1 and α2 subunit-containing GABA$_A$Rs with spectrin isoforms further, purified α1 and α2 receptors were resolved by BN-PAGE and immunoblotted

(Fig. 4a). Coomassie staining was used to confirm equal protein loading. Immunoprecipitated α1 and α2 receptors showed immunopositivity for α1. Immunopurified α1 receptors showed a distinct band of α1 immunopositivity at 250 kDa and 720 kDa. Immunopurified α2 receptors showed α2 immunopositivity at around 270 kDa and 750 kDa – consistent with the 27 kDa pHluorin-myc modification on the α2 subunit. The mass-shifted 270 kDa α2 species was also immunopositive for the α1 subunit, further confirming the presence of mixed α2-α1 receptors. Both the α1 720 kDa band, and the α2 750 kDa band were immunopositive for β2 spectrin, however more β2 spectrin was observed with purified α1 receptors. In contrast, β4 spectrin was only observed in the α2 750 kDa band and was absent from the α1 720 kDa band. The observed band at 600 kDa is likely to be either a non-specific band or a protein complex containing Sptbn4 spectrin that is not associated with either GABA$_A$R subtype at the point of protein complex resolution by BN-PAGE, as it does not resolve at the same molecular weight as either GABA$_A$R subtype. β2 spectrin and β4 spectrin were not observed at lower molecular weights, confirming their protein complexes with GABA$_A$Rs remained intact.

Molecules that regulate the membrane trafficking and synaptic accumulation of GABA$_A$Rs mediate their effects via binding to the major intracellular domains of receptor subunits. Thus, we examine if these domains are capable of binding spectrins when expressed as glutathione-S transferase fusion proteins (GST). To test for possible interactions with spectrin isoforms the GST fusion proteins encoding the intracellular domains of the α1, α2, and α4 subunits (along with GST alone as a control) were exposed to murine purified plasma membrane lysates and bound material was eluted, resolved by SDS-PAGE and immunoblotted (Fig. 4b). β2 spectrin was detected binding to the intracellular domain of the α1 subunit but significantly less to α2 subunit ($p < 0.000648278$) and the α4 subunit ($p < 0.00126715$). No binding was observed to GST alone. In contrast, β4 spectrin specifically bound to the α2 subunit compared to the α1 subunit ($p = 0.020606747$), with none associated with α4 or GST alone.

In addition to the in vitro binding studies, we analyzed the subcellular distribution of β2 spectrin, β4 spectrin and synaptic GABA$_A$Rs in cultured neurons from pHα2 mice (Fig. 4c). α1 and α2 were found at inhibitory synapses on dendrites, marked by opposition to the inhibitory presynaptic marker protein; VGAT. Most of these dendritic puncta also contained β2 spectrin immunoreactivity. In mature neurons β4 spectrin is highly enriched in the AIS and, consistent with this, immunoreactivity for β4 spectrin was limited to this structure in our cultures. In these structures puncta of GABA$_A$Rs containing both the α1 and α2 subunits also contained β4 spectrin immunoreactivity.

Taken together, these results suggest that β2 spectrin and β4 spectrin are capable of selective binding to the intracellular domains of the GABA$_A$R α1 and α2 subunits, respectively. They also suggest that β2 spectrin is found at or close to inhibitory synapses on dendrites, while β4 spectrin is associated with GABA$_A$Rs on the AIS.

**Examining the role that β2 spectrin plays in regulating GABA$_A$R accumulation at synapses.** To evaluate the importance of spectrin localization at inhibitory synapses we prepared primary cultured neurons from SPTBN1 floxed mice (SPTBN1$^{Flox}$). At DIV3, the cultures were infected with adeno-associated viruses (AAVs) expressing fluorescent GFP and *Cre* recombinase under the control of the CaMKIIα promoter (AAV9-CaMKII-eGFP-*Cre*; AAV-*Cre*) or a virus expressing GFP alone (AAV9-CaMKII-eGFP; AAV-GFP). The CaMKII promoter was used to restrict expression of the respective transgenes to principal neurons. The

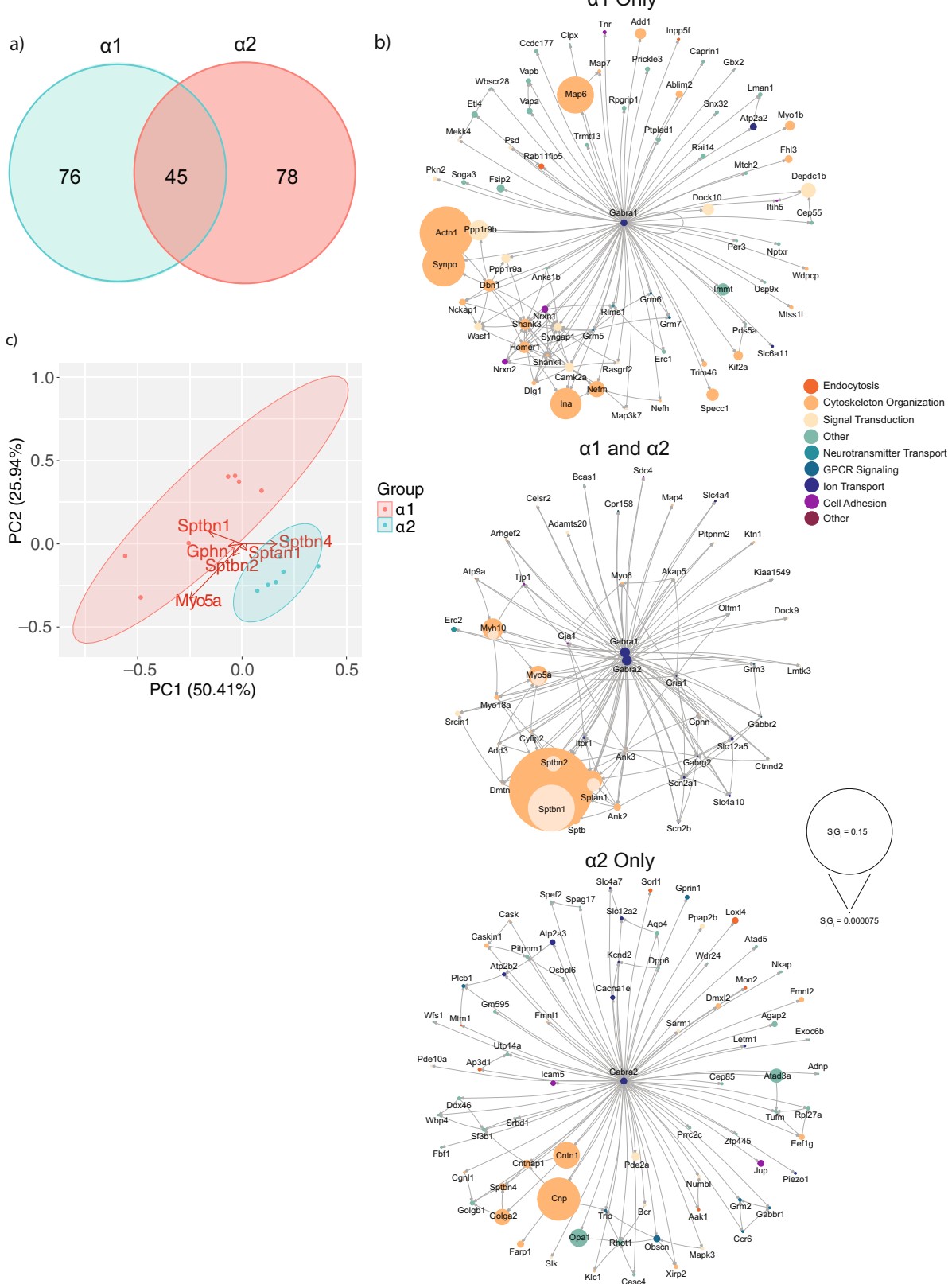

AAV-GFP and AAV-Cre viral infection allowed us to study conditional β2 spectrin$^{-/-}$ and β2 spectrin$^{+/+}$ cells respectively. At DIV18-21, neurons were lysed, and subject to immunoblotting (Fig. 5a). β2 spectrin levels were significantly reduced in β2 spectrin$^{-/-}$ neurons to approximately 50% of control ($p = 0.000052$). Interestingly, β4 spectrin levels were significantly increased ($p = 0.024824603$) to

approximately 150% of control. This result agrees with previous studies suggesting a reciprocal relationship between the expression levels of these distinct spectrin isoforms[20]. In contrast, the levels of α2 spectrin, a core component of spectrin heterodimers, was unaltered ($p = 0.110899224$) along with the expression of Gphn ($p = 0.74196358$) and VGAT ($p = 0.091484159$). Likewise, the

**Fig. 3 Comparing the proteomes associated with GABA$_A$R subtypes.** a) α1 and α2 containing protein complexes were immunopurified from mouse forebrain plasma membrane fractions. The complexes were resolved by BN-PAGE and the high molecular weight complexes were identified (~700 kDa). These were excised and the proteins identified by label-free quantitative proteomics. The identified proteins in α1 and α2 complexes were compared to control samples, and those proteins significantly enriched were included for downstream analysis. A Venn diagram showing the number of significant proteins unique to α1 or α2 and overlapping proteins ($n = 7$). The raw data are contained in Supplementary Data 5. b) The lists of significantly enriched unique and overlapping proteins were used to create network diagrams. Known interactions between detected proteins were obtained using stringent, high confidence, direct experimental association parameters from STRINGdb. These were used to construct a network diagram of protein nodes and arrows to indicate known interactions. The node diameter was scaled relative to the $S_iG_i$ values detected for each protein. An overlay of Gene Ontology (GO) Biological Process terms was used to provide protein classification information ($n = 7$). c) PCA analysis of each biological replicate for α1 and α2 containing protein complexes. PCA loadings were also included to show the contribution of each protein to the position of samples on the PCA plot ($n = 7$).

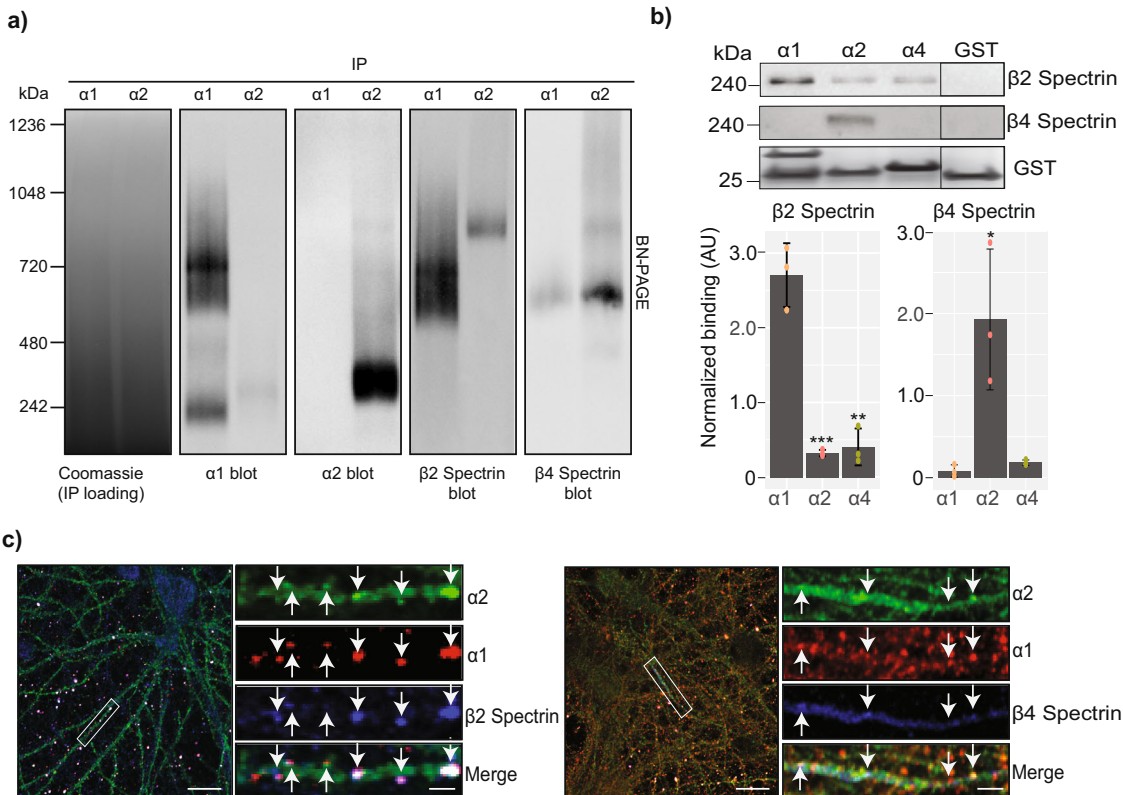

**Fig. 4 Assessing the association of spectrins with GABA$_A$Rs.** a) Immunopurified α1 and α2 containing GABA$_A$Rs were resolved by BN-PAGE, transferred, and immunoblotted for α1 and pHα2 subunits to visualize their high molecular weight complexes. These were also probed for the presence of β2 and β4 spectrin to demonstrate that they are also present in complexes with α1 and α2 containing GABA$_A$Rs. Representative images from an $n = 3$ experiment. b) GST-fusion proteins for intracellular domains of the α1, α2, and α4 were created along with GST alone. These were used to perform pulldowns from brain plasma membrane lysates and probe for β2 spectrin and β4 spectrin interaction by immunoblot. Equal loading of the fusion proteins was confirmed by colloidal Coomassie staining. Representative images from an $n = 3$ experiment. The amount of β2 spectrin and β4 spectrin pulled down was quantified by densitometry ($n = 3$) (*$p < 0.05$, **$p < 0.01$, ***$p < 0.001$). The raw data are contained in Supplementary Data 6. c) Immunocytochemistry for α1, α2, β2 spectrin, and β4 spectrin was carried out in fixed and permeabilized DIV 21 primary cultured pHα2 mouse neurons. Dendritic and AIS regions were imaged by confocal microscopy. Representative images from $n = 16$ from 4 individual cultures. Scale bar = 10 μm and 2 μm in cropped images. Error bars represent the SEM. The raw membrane/gel images are shown in Supplementary Figure 3.

expression levels of the GABA$_A$R α1 ($p = 0.14525492$) and α2 ($p = 0.565935661$) subunits were unaffected by reducing β2 spectrin expression.

We also measured how the modifications in β2 spectrin and β4 spectrin effected inhibitory synapse density using immunos-taining. DIV 18–21 β2 spectrin$^{-/-}$ and β2 spectrin$^{+/+}$ cells were fixed, permeabilized and immunostained for VGAT and Gephyrin. Inhibitory synapses were defined by VGAT positivity opposing Gephyrin positivity. The dendritic synapse density was significantly reduced ($p = 0.0000054726$) by approximately 50% in β2 spectrin$^{-/-}$ neurons (Fig. 5b).

**Ablating β2 spectrin expression differentially impacts GABA$_A$R accumulation on the AIS and dendrites.** Having established that β2 spectrin and β4 spectrin preferentially associate with the α1 and α2 GABA$_A$R subunits, respectively, we studied the effect of reducing β2 spectrin expression, and the observed compensatory increase in β4 spectrin expression on the density of α1 and α2 GABA$_A$R puncta in the dendrites and AIS of primary cultured neurons (Fig. 6a, b). On the AIS, β2 spectrin$^{-/-}$ neurons had a significantly lower ($p = 0.000084651$) density of approximately 50% α1 puncta compared to β2 spectrin$^{+/+}$ neu-rons. This was accompanied by a significant increase ($p =$

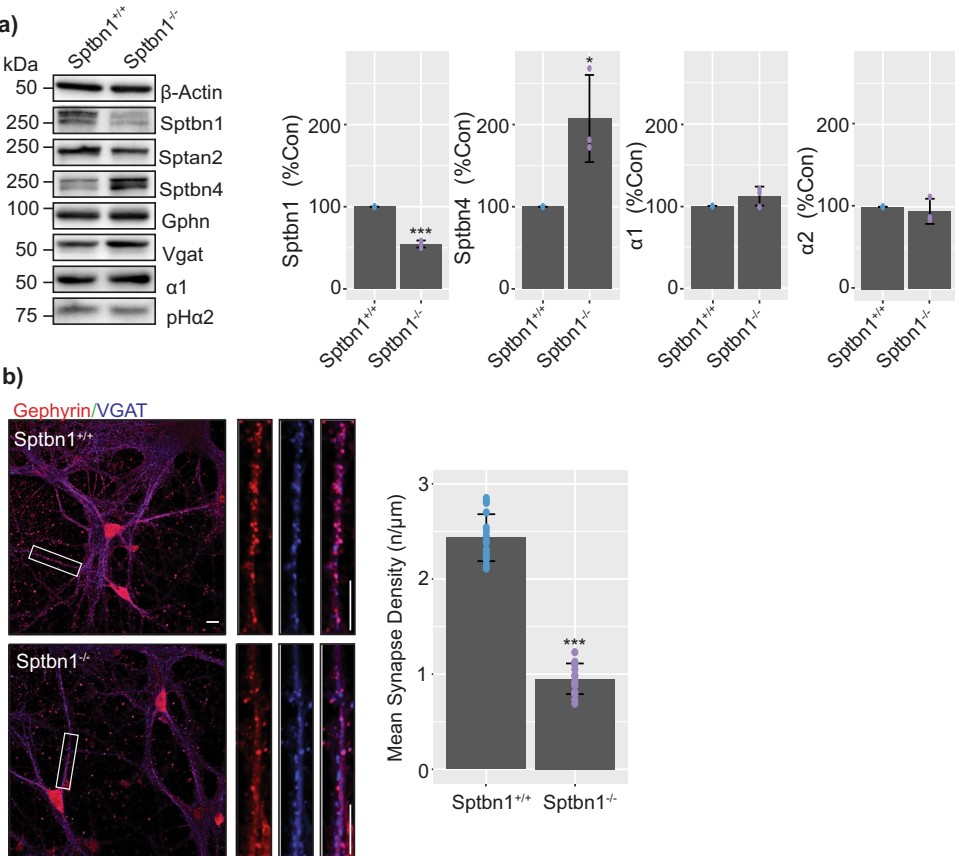

**Fig. 5 Determining the role that SPBTN1 plays in regulating GABA$_A$R expression levels and synaptic accumulation.** a) Primary cultured pHα2xSPTBN1$^{Flox}$ mouse neurons were infected with either AAV-GFP or AAV-GFP-Cre to create nuclear-restricted GFP-labelled β2 spectrin$^{+/+}$ or β2 spectrin$^{-/-}$ cultures respectively. At DIV 21 the cells were lysed, the proteins resolved by SDS-PAGE, and immunoblots carried out. Densitometry was carried out to quantify the differences in expression of key proteins between β2 spectrin$^{+/+}$ or β2 spectrin$^{-/-}$ cultures (Representative images from $n = 3$, $n = 3$ for densitometry analysis *$p < 0.05$, ***$p < 0.001$). b) Representative images of β2 spectrin$^{+/+}$ and β2 spectrin$^{-/-}$ DIV21 primary cultured neurons, fixed, permeabilized, and immunostained for VGAT and Gephyrin. Scale bar = 10 μm and 2 μm in cropped images. Synapses were determined by VGAT and Gephyrin colocalization, counted, and normalized to the length of the measured processes ($n = 16$ from 4 individual cultures, ***$p < 0.001$). Error bars represent the SEM. The raw data are contained in Supplementary Data 7. The raw membrane/gel images are shown in Supplementary Figure 4.

0.00012195) in the density of α2 puncta of approximately 50%. There was a similar significant reduction ($p = 0.00040454$) in α1 puncta in the dendritic compartment of β2 spectrin$^{-/-}$ neurons, where they were reduced by approximately 60%. This was again accompanied by a significant ($p = 0.00053214$) 50% increase in the density of α2 puncta.

Taken together, these results suggest that reducing β2 spectrin expression, which subsequently increases β4 spectrin expression, specifically reduces the synaptic accumulation of the α1 subunit containing GABA$_A$Rs and increases the accumulation of the α2 subunit containing GABA$_A$Rs in both the AIS and dendritic compartments.

**Reducing β2 spectrin expression modifies the amplitude of inhibitory synaptic currents.** Our biochemical and immunostaining experiments revealed modifications in the clustering of GABA$_A$Rs consisting of different α subunits on dendrites and on the AIS in neurons deficient in β2 spectrin. To determine if these modifications impacted the inhibitory synaptic signaling we compared the properties of miniature inhibitory synaptic currents (mIPSCs) in primary neurons cultured from SPTBN1$^{Flox}$ mice infected with AAVs expressing *Cre* recombinase (Fig. 7a). Compared to control cells, mIPSC average peak amplitudes were increased in *Cre* expressing neurons ($p = 0.03734$) while their frequency and decay rates were unaltered (Fig. 7b). Consistent

with the increase in mIPSC average peak amplitudes, we observed a pronounced rightward shift in the cumulative distribution of mIPSC amplitude (Fig. 7c) ($p = 0.02556$). The difference in amplitude between β2 spectrin$^{-/-}$ and β2 spectrin$^{+/+}$ neurons complements our previous results showing alterations in the balance of GABA$_A$R α subunit expression in β2 spectrin$^{-/-}$ neurons. Our results indicate that reduction of β2 spectrin expression in neurons significantly increases the amplitude of inhibitory synaptic currents.

## Discussion

Neurons have the capacity to differentially target specific GABA$_A$R subtypes to dendritic and axo-axonic synapses. Immunolocalization studies suggest that receptors containing the α1 subunit are predominantly found at dendritic synapses, while the α2 subunit is enriched at their equivalents on the axon initial segment (AIS)[5,6,21].

To explore the underlying mechanisms further, we used immunopurification, coupled with BN-PAGE to isolate native populations of GABA$_A$Rs assembled from α1 and α2 subunits. Using quantitative mass spectroscopy their subunit composition and the components of their associated proteomes were compared. Our results revealed that the majority of the α1 subunit co-purified with the β1-3 and γ2 subunits. Lower, but significant levels of the α2 and α3 subunits were also detected. Consistent

## a)

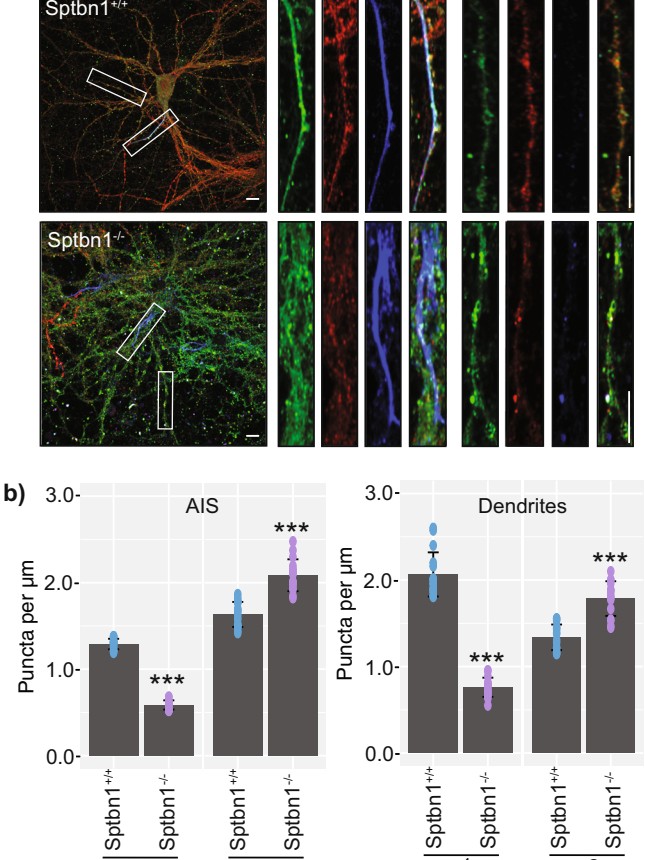

## b)

**Fig. 6 Examining the effects of ablating β2 spectrin expression on GABA<sub>A</sub>R accumulation at the AIS and dendrites.** a) Representative images of DIV21 β2 spectrin$^{+/+}$ and β2 spectrin$^{-/-}$ neuronal cultures, fixed, permeabilized and immunostained for α1, α2, and AnkG. The AIS and dendritic regions were determined by AnkG positivity or negativity respectively. Scale bar = 10 μm and 2 μm in cropped images (n = 16 from 4 individual cultures). b) Puncta of α1 and α2 were counted in the AIS and dendritic regions of interest and normalized to the length of the measured processes (n = 16 from 4 individual cultures, ***p < 0.001). Error bars represent the SEM. The raw data are contained in Supplementary Data 8.

with this, comparable levels of the α1 subunit co-purified with the α2 subunit. Thus, neurons express a large population GABA<sub>A</sub>Rs containing α1 together with the β1-3 and γ2 subunits[4,22]. They further suggest that the majority of α2 receptors also contain the α1 subunit. Consistent with our biochemical studies, immunostaining revealed that receptors containing both the α1 and α2 subunits were enriched on the AIS, while those containing just α1 subunit were predominantly found within dendrites. The existence of α2α1 receptors expressed in high abundance has multi-faceted implications. Depending on the subunit arrangement, GABA and benzodiazepine sensitivity could be altered relative to α1/α1 or α2/α2 receptors. The presence of the α1 subunit would also confer higher gephyrin affinity to α2-containing receptors, potentially altering their synaptic stability[10,23].

To gain insights into the mechanisms that orchestrate the subcellular targeting of GABA<sub>A</sub>Rs, the proteins that co-purified with these specific subtypes were compared. We identified sets of highly reproducible α1 and α2 receptor-binding proteins that including known GABA<sub>A</sub>R interactors, such as gephyrin[24], and

Cyfip2[25]. High levels of spectrins were associated with α1 and α2 receptor subtypes as high molecular mass complexes. In neurons, spectrin tetramers consisting of α2 spectrin and a variable β spectrin that create a sub-membranous lattice that interacts with the actin cytoskeleton. As measured by LC-MS/MS, α2 spectrin was co-purified with both GABA<sub>A</sub>R subtypes while β2 spectrin was enriched with those containing α1 subunits. In contrast, β4 spectrin was only associated with GABA<sub>A</sub>Rs containing the α2 subunit. In vitro binding demonstrated that β2 spectrin preferentially bound to the ICD of the α1 subunit, while β4 spectrin specifically bound to the corresponding region of the α2-ICD. β4 spectrin is enriched at the AIS and has been shown to be recruited to this structure by AnkG which also co-purified with GABA<sub>A</sub>Rs. Consistent with this the GABA<sub>A</sub>R α2 subunit was enriched on the AIS with β4 spectrin. In contrast, β2 spectrin was enriched at dendritic inhibitory synapses together with the α1 subunit. Further studies will be required to determine if these interactions are direct or via an intermediate, such as gephyrin, collybistin or GABARAP[26].

To assess the importance of spectrins for the formation of inhibitory synapses we used neurons in which β2 spectrin expression had been conditionally ablated. As previously observed, neurons lacking β2 spectrin showed a compensatory increase in β4 spectrin expression (Fig. 5), with a more dispersed staining pattern and a longer, more disorganized AIS[20]. While the balance of β spectrin isoforms was disrupted, no change was observed in the expression of the α spectrin isoform; α2 spectrin, or the expression of α1 or α2 GABA<sub>A</sub>R subunits. Decreasing β2 spectrin levels reduced the number of dendritic synapses containing the α1 GABA<sub>A</sub>R subunit. In contrast α2 subunit puncta on the AIS and dendrites were increased. As no change in the total expression of the α1 or α2 GABA<sub>A</sub>R subunits were observed, the disruption of β spectrin balance did not affect gross GABA<sub>A</sub>R subunit expression. Rather, the synapse targeting changes observed in α1 and α2 GABA<sub>A</sub>Rs could be the function of one or several cellular processes, including altered endocytosis[27], exocytosis[28], or synapse confinement directly, similar to the molecular functions of gephyrin[29]. This redistribution of α1 and α2 GABA<sub>A</sub>R synapse targeting resulted in changes in neuronal electrophysiological properties. Whole-cell recording, using seals on the cell body are biased for events on the soma and AIS due to space clamp limitations[30]. Therefore, the demonstrated increase in the amplitude of inhibitory synaptic currents is likely to a product of increased α2 GABA<sub>A</sub>R expression on the AIS.

Collectively our studies demonstrate an integral role for spectrins in orchestrating the synapse-specific targeting of GABA<sub>A</sub>Rs, a process that determines the efficacy of fast neuronal inhibition. Spectrinopathies have debilitating neurological features[31,32], often characterized by refractory seizures and epilepsies[33,34]. Therefore, these phenotypes may arise in part from deficits in the formation of inhibitory synapses.

## Methods

**Animals.** Animal studies were performed according to protocols approved by the Institutional Animal Care and Use Committee (IACUC) of Tufts Medical Center. 8–12-week-old C57BL/6 male and female mice were kept on a 12 h light/dark cycle with ad libitum access to food and water. Transgenic mice expressing the phluorin-myc n-terminal modification of the α2 subunit (pHα2) were created as previously described[11].

**Antibodies.** The following antibodies were used for immunoprecipitation (IP), immunoblot (IB) or immunocytochemistry (ICC): Gabra1 (IP, IB – 1:1000, mouse, Antibodies Inc 75–136), myc (IP, mouse, Thermo Scientific PIER88843), GFP (IB – 1:1000, rabbit, Cell Signaling 2956), GFP (ICC – 1:1000, chicken, Abcam ab13970), AnkG (ICC – 1:1000, rabbit, SYSY 386003), VGAT (IB - 1:1000, ICC - 1:1000, guinea pig, SYSY 131004), β2 spectrin (IB – 1:1000, ICC0 - 1:500, mouse, Sigma-Aldrich SAB4200662), β4 spectrin (IB – 1:1000, ICC – 1:500, mouse, Antibodies

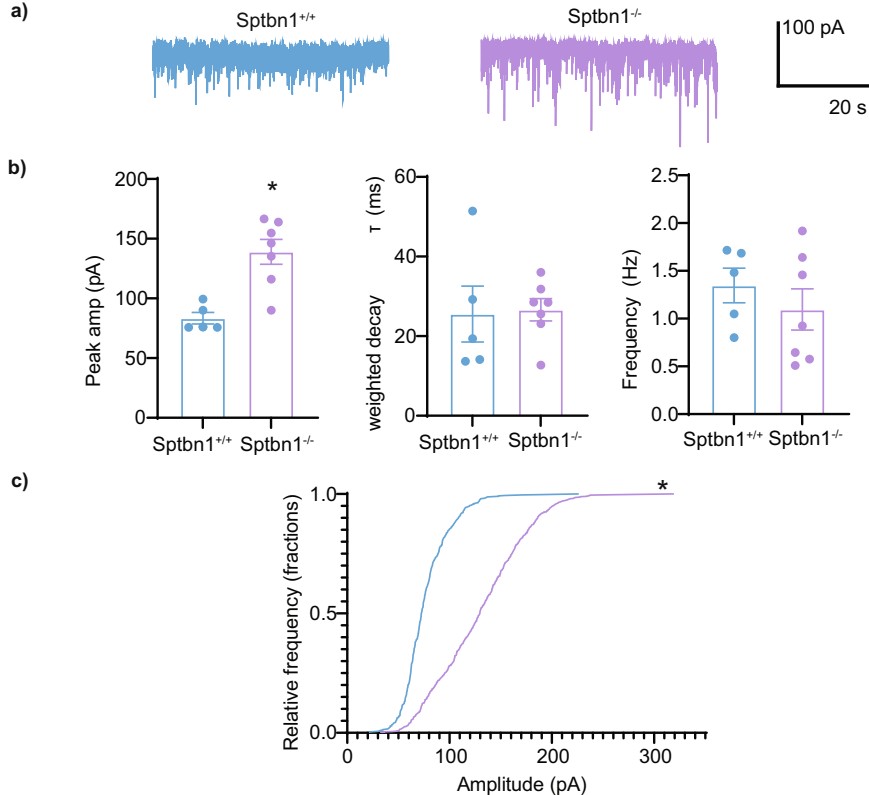

**Fig. 7 Determining the effects of β2 spectrin ablation on the properties of inhibitory synaptic currents.** a) Representative mIPSCs from DIV 18–22 neurons cultured from SPTBN1$^{Flox}$ mice infected with AAV-GFP (black) or AAV-GFP-Cre (red) to create Sptbn1$^{+/+}$ or Sptbn1$^{-/-}$ cultures respectively. b) Bar graphs show average mIPSC peak amplitude (pA), weighted decay tau (ms), and frequency (Hz). Only mIPSC amplitude was significantly larger in Sptbn1$^{-/-}$ neurons (**, significantly different from control, P = 0.03734, $n$ = 5–7 cells). c) Frequency distribution of mIPSC events of different amplitudes ($p = 0.02556$; $n = 5$–7 cells). Error bars represent the SEM. The raw data are contained in Supplementary Data 9.

---

Inc 75–377), Gabra1 (ICC – 1:1000, rabbit, Abcam ab33299), β-actin (IB - 1:5000, mouse, Sigma-Aldrich A1978), α2 spectrin (IB – 1:1000, rabbit, Cell Signaling 2122), Gephyrin (IB – 1:1000, rabbit, Cell Signaling 14304).

**Plasma membrane isolation**. Plasma membrane isolation was carried out as previously described[14,35]. Briefly, murine forebrain (from 7 mice) was isolated in dissection buffer (225 mM mannitol, 75 mM sucrose, 30 mM Tris–HCl pH 7.4) on ice. The tissue was transferred to homogenization buffer stored on ice containing; 225 mM mannitol, 75 mM sucrose, 0.5% (wt/vol) BSA, 0.5 mM EGTA, 30 mM Tris–HCl, pH 7.4 supplemented with mini cOmplete protease inhibitor and PhosSTOP. The brains were homogenized using 14 strokes of a Dounce homogenizer. All the following centrifugation steps were carried out at 4 °C. The samples were initially centrifuged at 800 × $g$ for 5 min to facilitate the removal of nuclei and non-lysed cells. After discarding the pellet, the supernatant was spun again at 800 × $g$ for 5 min to remove residual nuclei and non-lysed cells. The supernatant then centrifuged at 10,000 × $g$ for 10 min to remove mitochondria. The pellet was discarded, and the supernatant spun again at 10,000 × $g$ for 10 min to remove mitochondrial contamination. The plasma membrane fraction was then pelleted at 25,000 × $g$ for 20 min. Following resuspension in starting buffer, the plasma membrane fraction was spun again at 25,000 × $g$ for 20 min to remove cytosolic and ER/Golgi contamination.

**Immunoprecipitation**. Immunoprecipitations (IPs) were carried out as previously described[14]. Briefly, anti-c-myc beads (Thermo Fisher Scientific) or Protein G Dynabeads (Thermo Fisher Scientific) were washed three times with phosphate-buffered saline with 0.05% Tween-20 (PBS-Tween) and resuspended in PBS-Tween containing non-immune IgG or with antibodies against the α1 subunit respectively. The antibody concentration was experimentally predetermined (Supplementary Figure 1). For α1 IPs, the antibody was crosslinked onto the beads by washing twice with 0.2 M triethanolamine (pH 8.2) (TEA), and then incubated for 30 min with 40 mM dimethyl pimelimidate (DMP) in TEA at room temperature. The beads were transferred to 50 mM Tris (pH 7.5) and incubated at room temperature for a further 15 min. The beads were washed three times with PBS-Tween and resuspended in solubilized plasma membranes in ice-cold Triton lysis buffer, supplemented with mini cOmplete protease inhibitor and PhosSTOP. The

immunoprecipitation reaction was incubated overnight at 4 °C. The beads were then washed three times with PBS-Tween and eluted either with 2x sample buffer (for SDS-PAGE) or soft elution buffer [0.2% (wt/vol) SDS, 0.1% Tween-20, 50 mM Tris–HCl, pH = 8.0][12] (for BN-PAGE).

**GST-Fusion protein pulldown**. GST-fusion proteins for the intracellular domains (ICDs) of α1, α2, and α4 were synthesized in E. coli and immobilized on magnetic glutathione beads (Thermo Fisher Scientific)[36]. These were washed 3 times with PBS-Tween and incubated with plasma membrane fractions solubilized in Triton lysis buffer overnight at 4 °C. The beads were washed 3 times in 0.05% PBS-Tween and the proteins eluted in 2x sample buffer.

**BN-PAGE**. BN-PAGE was carried out as previously described[14]. Briefly, protein samples were eluted in soft elution buffer and 4x NativePAGE sample buffer and G250 additive (Thermo Fisher Scientific) were added. Samples were loaded onto NativePAGE gradient gels (4–16%) (Thermo Fisher Scientific). Gels were run for 2–3 h followed by preparation for Coomassie staining or immunoblotting.

**SDS-PAGE**. SDS-PAGE was carried out as previously described[37]. Briefly, protein Bradford assays (Bio-Rad, Hercules, CA, United States) were carried out to measure and subsequently normalize protein concentration. Samples were diluted in 2x sample buffer and 5–100 μg of protein was loaded onto a 7–15% tris-glycine polyacrylamide gel depending on the molecular mass and relative abundance of the target protein.

**Coomassie staining**. The gels were sealed throughout this process to prevent contamination. Gels containing resolved proteins were fixed in 50% ethanol and 10% acetic acid, washed in 30% ethanol, washed in molecular grade ultrapure water, then stained with EZ blue stain at room temperature for 2–24 h. The gels were destained in molecular grade ultrapure water and imaged using a ChemiDoc MP (Bio-Rad). Bands of interest were excised for liquid chromatography-tandem mass spectrometry (LC-MS/MS).

**Immunoblotting**. Following BN-PAGE or SDS-PAGE, proteins were transferred to PVDF or nitrocellulose membranes respectively. PVDF membranes were fixed in 8% acetic acid, washed with molecular grade ultrapure water, air-dried, and destained with 100% methanol. PVDF and nitrocellulose membranes were prepared for immunoblotting in the same way from this point onwards. Membranes were blocked in 5% milk in tris-buffered saline 0.1% Tween-20 (TBS-T) for 1 h, washed with TBS-T, and then probed with primary antibodies prepared in TBS-T overnight (see the antibodies section for dilution information). The membranes were washed and incubated for 1 h at room temperature with HRP-conjugated secondary antibodies (1:5000 – Jackson ImmunoResearch Laboratories, West Grove, PA, United States) and imaged using a ChemiDoc MP (Bio-Rad). Band intensity was compared to α-tubulin or β-actin as a loading control. Resolved protein bands from raw images were analyzed Fiji. Where possible, biological replicates were run on the same gels for comparison, and the area under the curve was calculated for each band. Average signal and standard error of the mean (SEM) were calculated for each treatment group and ANOVA or t-test carried out as appropriate using R for statistical comparison of protein expression.

**Primary neuron culture**. Mixed cortical and hippocampal primary neuron culture was carried out as previously described[38]. Briefly, P0 mice were anesthetized on ice and the brains removed and dissected in Hank's buffered salt solution (HBSS) (Thermo Fisher Scientific) supplemented with 10 mM HEPES. The dissected tissue was trypsinized and triturated to dissociate the neurons. Viable cells were counted using a hemocytometer and trypan blue staining, followed by plated on pre-coated poly-l-lysine-coated 13 mm coverslips in 24-well plate wells at a density of $2 \times 10^5$ cells/ml in Neurobasal media (Thermo Fisher Scientific). *Cre*-mediated knockout of *SPTBN1* in *SPTBN1^Flox^* neurons was carried out using AAVs[39]. Briefly, at days in vitro (DIV) 3, $10^5$ genomic copies per cell of CamKII-AAV9-GFP (Addgene, Watertown, MA, United States) were added to the neuronal media. After 24 h, the media was replaced with conditioned media. The cells were harvested for ICC or IB at DIV18-21.

**Immunocytochemistry**. Immunocytochemistry was carried out as previously described[39]. Briefly, DIV18-21 primary cultured neurons were fixed for 10 minutes in 4% paraformaldehyde (PFA) in PBS. The cells were permeabilized for 1 h in blocking solution – 50:50 mixture of 5% BSA in PBS and normal goat serum (NGS). The cells were probed with primary and then fluorophore-conjugated secondary antibodies (Alexa Fluor 488, 555, and 647; Thermo Fisher Scientific, 405 Dylight; Jackson ImmunoResearch) diluted in blocking solution for 1 h each at room temperature (see antibodies section for dilution information). The coverslips were washed in PBS, dried, and mounted onto microscope slides with Fluoromount-G (SouthernBiotech, Birmingham, AL, United States). The samples were imaged using a Nikon Eclipse Ti (Nikon Instruments, Melville, NY, United States) confocal microscope using a 60x oil immersion objective lens. Synapse counts and colocalization studies were carried out as previously described[40]. Briefly, analysis was performed using the Synapse Counter plugin in the FIJI software package[16]. 1024 × 1024 images were auto-thresholded using the Otsu Thresholding method. The rolling ball radius (background subtraction) and maximum filter parameters were set to 7 and 1, respectively. Default colocalization settings were used that accept 33–100% overlap between pre- (VGAT) and post-synaptic (Gephyrin) markers. Average overlap and standard error of the mean (SEM) were calculated for each treatment group and ANOVA/t-test carried out as appropriate using R for statistical comparison.

**Electrophysiology**. Whole-cell recordings were conducted at 32 °C in bath saline solution (140 mM NaCl, 2.5 mM KCl, 2 mM CaCl₂, 1.5 mM MgCl₂, 10 mM HEPES, 11 mM glucose, pH to 7.4 with NaOH). To examine miniature inhibitory postsynaptic currents (mIPSCs), 0.3 μM tetrodotoxin (TTX) was added to the bath solution and cells continuously perfused with DNQX (20 μM, 6,7-dinitroquinoxaline-2,3-dione) and AP5 (50 μM dl-2-amino-5-phosphonopentanoic acid) to block glutamatergic excitatory synaptic transmission. Recording pipettes (5-7 MΩ) were pulled from borosilicate glass (World Precision Instruments) and filled with internal solution (140 mM CsCl, 1 mM MgCl₂, 0.1 mM EGTA, 10 mM HEPES, 2 mM Mg-ATP, 4 mM NaCl, 0.3 mM Na-GTP, pH to 7.2 with CsOH). Recordings of mIPSCs were obtained after a 5-minute stabilization period upon establishing a whole-cell configuration. Current recordings were low-pass filtered at 2 kHz with an Axopatch 200B amplifier (Molecular Devices) and analyzed with Clampex 10 software (Molecular Devices). Cells were omitted from analysis if their access or series resistances changed by >20%. Individual mIPSC events were visually inspected and were accepted based on having a stable baseline, sharp rising phase, and a single peak. Only recordings with a minimum of 200 events fitting these criteria were used for analysis. mIPSC amplitude and frequency from each experimental condition were combined and expressed as mean ± SEM. mIPSC decay times were averaged from 100 consecutive events, fit to a double exponential, and weighted decay tau (τ) obtained. Statistical analysis for average mIPSC kinetics was performed using the unpaired T-test where $p < 0.05$ is considered significant. mIPSC amplitudes were fitted with a Gaussian function:

**Equation 1**. Gaussian function used to fit mIPSC amplitudes.

$$f(x) = \sum_{i=1}^{n} A_i \frac{e^{-(x-\mu_i)^2/2\sigma_i^2}}{\sigma_i \sqrt{2\pi}} + C$$

For $n$ components, the fit solves for the amplitude $A$, the Gaussian mean amplitude current $\mu$, the Gaussian standard deviation $\sigma$ and the constant y-offset $C$ for each component $i$.

The experimental numbers used were determined from the following power analysis based on actual data from previous experiments (Nakamura et al., 2016, Nathanson et al., 2019 Kontou et al., 2021); Difference in Means = 0.400, Standard Deviation = 0.1500, Power = 0.950, Alpha = 0.0500 demonstrating that 4 replicates of each group are required to reach statistical significance.

**Protein Analysis by LC-MS/MS**. Quantitative label-free proteomic analysis was carried out as previously described[41]. The gel bands of interest were excised and cut into 1 mm³ pieces. Following in-gel trypsin digestion[42], the gel pieces were washed and dehydrated with acetonitrile for 10 min and dried in a speed-vac. The gel pieces were then rehydrated with 50 mM ammonium bicarbonate solution containing 12.5 ng/μl modified sequencing-grade trypsin (Promega, Madison, WI, United States) and incubated for 45 min at 4 °C. The trypsin solution was removed, and 50 mM ammonium bicarbonate solution added and incubated at 37 °C overnight. The peptides were extracted by washing with 50% acetonitrile and 1% formic acid and the extracts were dried in a speed-vac for 1 hour. Prior to analysis the samples were stored at 4 °C, then reconstituted in 5–10 μl of HPLC solvent A (2.5% acetonitrile, 0.1% formic acid). A nano-scale reverse-phase HPLC capillary column was created by packing 2.6 μm C18 spherical silica beads into a fused silica capillary (100 μm inner diameter × ~30 cm length) with a flame-drawn tip[43]. The column was equilibrated, and each sample was loaded using a Famos auto sampler (LC Packings, San Francisco, CA, United States) onto the column. A gradient was formed between solvent A, and increasing concentrations of solvent B (97.5% acetonitrile, 2.5% water, and 0.1% formic acid). Acquisition time was 16–79 min. As peptides were eluted, they were subjected to Nanospray ionization (NSI) and then entered an LTQ Orbitrap Velos Pro ion-trap mass spectrometer (Thermo Finnigan, San Jose, CA, United States). An MS1 value of 70k was used, with a scan range of m/z 85–2000, and charge-state screening parameters set to +2 to +5. A centroid acquisition mode was used, with a precursor ion isolation window of 2 m/z and 35% normalized collision energy. Eluting peptides were detected and the most intense were isolated using the Top 10 scan mode and fragmented by Higher-energy C-trap dissociation (HCD). An Orbitrap mass spectrometer with a resolution of 17.5 k and the dynamic exclusion settings was used to analyze MS2 ions, with a 30 s repeat duration, 60 s exclusion duration, $n = 1$, 10 ppm mass width, to produce a tandem mass spectrum of specific fragment ions for each peptide.

**Peptide/protein searches**. Raw MS data processing was carried out as previously described[14]. Peptide sequences were determined by matching protein or translated nucleotide database sequences with the acquired fragmentation pattern using the MSGF + [44]. Raw.mzXML files were used to interrogate the UniProt mouse reference proteome (last modified May 4th 2020, containing 21989 sequences) also containing with the Thermo list of common contaminants. The search was carried out using settings for high-resolution Orbitrap mass spectrometers, tryptic digestion, no limit to enzyme missed cleavages, 20 ppm precursor mass tolerance, charge states of +2 to +5, minimum and maximum peptide lengths of 6–40 amino acids in length, respectively, and a fixed modification of standard amino acids with carbamidomethyl (C + 57). Peptide identification was scored by MSGF + Q- (PSM-level target-decoy approach) and E- (expected number of peptides in a random database) scores. These were used for quality control in the initial protein screening process for associated proteins, and phospho-modified peptide screening[45].

**Proteomic data analysis**. The resulting .mzID files from the spectral searches were combined with .mzXML files using the MSnbase package in R (accessed July 20th 2020), and used to measure the spectral index normalized to global intensity ($S_IG_I$) for each protein. This has previously been shown to be an effective measurement for label-free protein quantification and to normalize replicate data[13], accounting for inter-run variability. The mass spectrometry proteomics data have been deposited in the ProteomeXchange Consortium databse via the PRIDE[46] partner repository with the dataset identifier PXD038703 and 10.6019/PXD038703. The $S_IG_i$ was calculated for each protein using all the of the detected peptides for each protein. After these individual features were combined, a Welch t-test was performed comparing the $S_IG_i$ for each protein detected by MS that immunoprecipitated with the α1 or α2 receptors compared to the $S_IG_i$ in their equivalent control. For the α1 receptor IP, the control was non-immune IgG bound to protein G Dynabeads. For the α2 receptor IP, the control was anti-myc Dynabeads with wildtype tissue, rather than tissue from the pHα2 mouse. Significantly enriched proteins were those with a subsequent p value lower than 0.05. Venn diagrams were produced using the Vennerable package in R (accessed January 10th, 2019), and only significantly detected proteins in all repeats were considered for downstream analysis. The $S_IG_I$ values for proteins contained within each gel band were normalized by z-transformation and used for principal Component Analysis

(PCA), which was carried out using PCA functions in the ggfortify package in R (accessed January 10th, 2019). The protein lists were compared against the latest version of the STRINGdb database[18] to establish known interactions and annotations for each protein using only high confidence, experimental evidence. Functional protein information, specifically the highest scoring Biological Process Gene Ontology terms, were extracted for each protein using the "mygene" package in R (accessed July 29th, 2020). The interaction for each protein with α1 or α2 was imputed, and network diagrams were constructed in R using the igraph package and the nodes were scaled to the $S_IG_I$ values for each protein (accessed February 1st, 2019).

**Statistics and reproducibility**. The results are expressed as mean ± the standard error of the mean (SEM). Statistical comparisons were carried out between two groups using Student's $t$ test. For proteomic data comparisons, a Welch $t$ test was used as it is more robust for comparing groups with samples of unequal variance. All replicates are independent biological replicates. For proteomic experiments, $n = 4$–7 was used, where each n was derived from fractionated samples derived from 7 mice. For immunoblot experiments, $n = 3$–4 was used, where samples were either derived from fractionated samples derived from 7 mice, or from individual primary cultured neurons from 6–12 pups. For immunocytochemistry experiments, $n = 4$ individual cultures was used when quantifications were carried out with broad aspect, low magnification images. When high magnification images were required, $n = 16$ was used derived from individual viral exposures from at least 4 individual primary cultures.

**Reporting summary**. Further information on research design is available in the Nature Portfolio Reporting Summary linked to this article.

## Data availability
The proteomic datasets generated during and/or analyzed during the current study are available in the ProteomeXchange repository under the accession number PXD038703. Unedited Western blot images are contained in Supplementary Figures 1b and 2-4. Source data underlying figures are presented in Supplementary Data 1-9.

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

## Acknowledgements

The work was supported in part by NIH grants MH118263, NS108378, NS101888, NS103865, and NS111338. We would like to thank Harvard Taplin and the Yale NIDA proteomics facilities for aiding in the proteomic data development.

## Author contributions

J.L.S. and S.J.M. conceptualized the project and wrote the paper. J.L.S., N.C., J.N., C.C., A.L., S.C. performed experiments. J.D., C.B., C.Z. provided and managed the animal colonies. M.R. and P.D. edited the paper.

## Competing interests

The authors declare no competing interests.
