## [Peer Review File · Communications Biology]

Reviewers' comments:

Reviewer #1 (Remarks to the Author):

In this insightful manuscript "Spectrin isoforms mediate the selective accumulation of GABA_A receptors at somatodendritic and axo-axonic synapses" the authors contribute multiple important findings related to GABA_ARs, the main receptor for generating fast inhibition in the CNS. Due to the diverse functions and large number of subunits that can form GABA_AR heteropentamers, defining the mechanisms of subtype sub-cellular compartmentalization is an important area of research and potential therapeutic drug development. Proteomics was used here to reveal that the majority of α 2-containing receptors co-assemble with α 1 subunits, whereas α 1 subunits can form GABA_ARs as the sole α subunit. In addition, proteomic and bioinformatic analysis identified the unique interactomes for α 1, α 2, and the overlapping α 1 α 2 GABA_AR interactome. Building on the specific interactome IDs, biochemical and immunofluorescence approaches were used to delineate GABA_AR interactions with distinct isoforms of spectrin, the neuronal scaffolds that link transmembrane proteins to the cytoskeleton. β 2 spectrin containing complexes were shown to preferentially associate with α 1 GABA_ARs, while β 4 spectrin associated with α 2 GABA_AR subtypes enriched on the AIS (axon initial segment). These associations occurred via the respective intracellular regions of the α 1 and α 2 subunits. Next, AAV mediated knockdown of β 2 spectrin resulted in a decrease in pre and post synaptic components of inhibitory synapses, and a loss of α 1 puncta in the AIS and dendrites. Interestingly, β 4 spectrin total protein levels were increased, concurrent with an upregulation of α 2 puncta in the in AIS and dendrites. Intriguingly the net change of these results led to enhanced GABAergic inhibition as seen from an increase in mIPSC amplitude. The observed perturbations of inhibitory synapses with knockdown of β 2 spectrin suggest potential GABA_AR contributions to the debilitating neurological phenotypes of spectrinopathies and recently identified neurodevelopmental syndromes in pathogenic SPTBN1 patient variants.

This is a valuable and interesting manuscript; however minor but important revisions are needed to clarify the findings and conclusions prior to acceptance. In addition, more editorial proofing is recommended (beyond the typos listed below).

MAJOR COMMENTS

- In Figure 5, the *sptbn1* ^{-/-} vs *sptbn1* ^{+/+} western blot was used to quantify loss of β 2 spectrin and an increase in β 4 spectrin. Yet the quantification of the other bands including GABA_AR alpha 1 & alpha 2 subunits was not provided. This data should be provided, particularly as the GABA_AR alpha 1 subunit appears increased in the representative blot data, and the alpha 2 subunit is unchanged. This is an important point to resolve as the IF data in Fig 6 shows increased alpha2 in AIS and dendrites and a 50% or greater loss of alpha 1 in dendrites and AIS. Is it possible alpha1 containing GABA_ARs are just not stabilized at synapses and accumulates intracellularly? The increase in alpha 2 clusters in dendrites & AIS should be confirmed by surface biotinylation or membrane fractions. This would help also in resolving why there is a 50% loss of inhibitory synapses in *sptbn1* ^{-/-} neurons (Fig 5b) at the same time as the rather surprising result that GABAergic inhibition is increased (mIPSC amplitude increase, Fig 7).
- One would also predict that a large decrease in alpha 1 subunits should lead to a larger tau decay, however Fig 6 shows decay as unchanged. There should be a large reduction in zolpidem sensitivity of the mIPSCs. This additional ephys experiment to verify loss of alpha 1 or alternatively a biochemical experiment showing enhanced surface levels of alpha 2 would strengthen and refine the conclusion. In general there is insufficient discussion of the ephys results combined with data from Fig 5 and Fig 6.
- In its current form Figure 3 is not providing sufficient information. An excel supplementary table of the detected protein in the 3 different conditions should be provided, preferably with the assigned enriched GO terms that were used to generate Fig 3b. What was the criterion used for determining "significant enrichment" vs the IgG & 9e10 control background? : "Only those proteins significantly

enriched compared to control were included for downstream analysis. Using these criteria 121 proteins were detected co-purifying with the $\alpha 1$ subunit, while 123 were associated with $\alpha 2$, 45 of which were common to both purifications (Fig 3a)." Further description is needed in the methods, as none is currently provided.

- Insufficient statistical information: Several figure legends do not indicate n #s and this needs to be added. #s neurons from # cultures, and # experiments for IPs, etc. Except for Fig 4 that indicated n=4 for Fig 4b & representative images of 4 independent cultures (n=4) for 4c. Fig 5 only indicates n# for the western, but no info is provided for Fig 5b. Fig 7 provides sufficient info on n#.

MINOR COMMENTS

Results

It would be helpful to define pHa2 before you use it. Consider reordering/rearranging with the following sentence for clarity.

It would be helpful in evaluating the immunostaining if it was mentioned in the results that the neurons were permeabilized, so puncta are total dendritic alpha 1 & alpha 2, rather than surface clusters.

Provide references for "This result agrees with previous studies suggesting a reciprocal relationship between the expression levels of these distinct spectrins."

It would be helpful to the general reader to indicate that the SPTBN1 gene encodes the $\beta 2$ spectrin protein.

Fig 4A: what is the 600 kDa complex with B4 spectrin in both lanes

The following sentence needs to be revised: " $\alpha 2$ spectrin co-purified both GABAAR subtypes while $\beta 2$ spectrin was enriched with those containing $\alpha 1$ subunits." No $\alpha 2$ spectrin IP was done or blotting in an IP reaction. Perhaps it was identified by mass spec, in which case the sentence should be reworded " $\alpha 2$ spectrin was co-purified with both GABAAR subtypes" or alternatively this is a typo.

The authors should also mention the following reference: Cousin MA et al., Pathogenic SPTBN1 variants cause an autosomal dominant neurodevelopmental syndrome. Nature Genetics, 2021; DOI: 10.1038/s41588-021-00886-z

TYPOS

Abstract:

Line 6 , $\alpha 1$ subunits

Line 10, "enriched GABAAR", replace with enriched with the $\alpha 2$ subunit

Line 15, "targeting of GABAAR," add s = GABAARs

Intro paragraph 2,

" $\alpha(1-6) \beta(1-3)$," Insert , between (1-6) and β

"majority GABAARs" , insert of, majority of GABAARs

Results:

"functional", change to functionally

"Gfp", change to GFP (also in other areas of results section)

720kDa for $\beta 2$ -containing = presumably $\alpha 2$

"single $\alpha 1$ subunit isoforms (Fig 2c)." = replace $\alpha 1$ with $\alpha 1$

"Examining the role that $\beta 2$ spectrin plays" - make all heading text bold
"condition $\beta 2$ spectrin", change to conditional
" $\beta 2$ spectrin-/- neurons (Fig 4b)." = change to (Fig 5b)

"This was accompanied by a significant increase ($p < 0.05$) in the density of $\alpha 1$ puncta or approximately 50%", change to $\alpha 2$ puncta

"revealed modifications in clustering of GABAAR α subunit", change to α or alpha subunit
"Neurons have capacity" - change to Neurons have the capacity

Discussion:

"neurons have capacity", replace with neurons have the capacity

"to isolated native populations", replace with to isolate native...

"consisting of $\alpha 2$ spectrin and variable β spectrin that create" , replace with ... β spectrin, create

"Decreasing $\beta 2$ spectrin levels reduced the number of dendritic synapses containing the $\alpha 1$ subunit numbers containing GABAAR $\alpha 1$ subunits" , change to Decreasing $\beta 2$ spectrin levels reduced the number of dendritic synapses containing the GABAAR $\alpha 1$ subunits

"In contrast $\alpha 2$ subunit puncta on the AIS and dendrites assembled from $\alpha 2$ subunits.", I am not sure what the authors meant here.

Figure legends:

Fig 1 -There is a duplicate description for d)

Fig 4:"These were also probed for the presence of and $\beta 4$ spectrin to demonstrate that they are also present in complexes with $\alpha 1$ and $\alpha 2$ containing GABAARs." should be presence of and $\beta 2$ and $\beta 4$

Reviewer #2 (Remarks to the Author):

The manuscript by Joshua L Smalley et al. combines biochemical, bioinformatics, neuronal culture imaging, and electrophysiology approaches to investigate the different α subunits of GABAAR compositions both at somatodendritic and the AIS synapses, which generate a set of informative data for the specific field. They first performed the biochemical purification using different α subunits antibodies and analyzed the isoform composition through LC-MS and neuronal culture, showing the co-localization of $\alpha 1$ and $\alpha 2$ at synapses. Further analysis of the proteomics data and bioinformatics information, they showed a differential distribution of $\alpha 1$ and $\alpha 2$ subunit-containing complexes. Interestingly, different β spectrins, a family of cytoskeleton proteins, are enriched with distinct α subunit of GABAAR at dendritic synapses or AIS synapses. Using $\beta 2$ spectrin depleting neurons, the authors showed the effects of GABAAR expression and demonstrated the mIPSC was also affected. Overall, this manuscript provides interesting insights into the complex regulation of the subunits compositions for GABAAR, a primary fast inhibitory receptor at the CNS. However, there are quite a lot of concerns should be addressed to enhance the conclusions present here.

Major points:

1) In the title, the authors claim that spectrin isoforms mediate the selective accumulation of GABAAR. However, this is not directly demonstrated. The data showed different spectrin isoforms co-purified with different α subunits, but not necessarily to be the reason for the selection. The reviewer suggests a more suitable title that represents the actual data.

2) Introduction: there are many conclusive descriptions of previous studies, but no reference papers were cited. For example, "Immunolocalization studies have revealed that principal neurons within the hippocampus express $\alpha 1-3$, $\beta 1-3$ and $\gamma 2$ subunits...", which immunolocalization studies and how they performed the experiments should be at least cited or described. It is the same situation for other

paragraphs in the whole introduction part.

3) Spectrins: the spectrin proteins are always considered as the cytoskeleton proteins that form tetrameric filaments providing the mechanical support. The authors have also cited the review paper (PMID:31191255) by Dr. Rasband, who is one of the authors for the present study. However, the authors considered spectrins as "scaffold" proteins instead of cytoskeleton proteins, which cannot be accepted by the reviewer. Furthermore, the Ankyrin family proteins, always forming duo complex with spectrins and functioning to linking membrane proteins to spectrin-based cytoskeleton, should be recognized as "scaffold" proteins, not simply described as the marker of the AIS. Although the authors made a short description in the "Discussion" part, the reviewer think it is worthy to make it clear.

4) Fig. 1: page5, there is no descriptions or any references about how to make "knock-in mice in which pH sensitive GFP and the 9E10 epitope (pHa2) have been incorporate at the N-terminus of this subunit", thus making it quite confusing and less convincing for the readers to appreciate these data. It would also help if a picture/panel could be present for the design of these experiments.

Fig.1 panel b), d), and f), the statistics are confusing and not stated how to compare and calculate the p values as no data for the WT row is shown.

Also, no LC-MS raw data is provided to justify the reliability of the results for Fig. 1.

For Fig. 1 legend, the last panel should be f), not d).

The d) panel, there is no "***" and "****" at all, so the description should be moved to f).

5) Fig.4: panel a), what is the major band (between 480 and 720 kDa, lower than the 750 kDa band for $\beta 4$ spectrin) of the $\beta 4$ blot, $\alpha 2$ line picture?

Fig. 4 legend: a) I think there is a missing of $\beta 2$ in the fourth row; c) no scale bar length for the imaging on the right and no numbers are provided (also for Fig 2).

6) Fig 5: panel a), it is quite strange for the statistical presentation for only one single column. Can the authors modify the comparison method to better present the results?

7) The authors showed that reducing $\beta 2$ spectrin expression results in reducing the synaptic accumulation of the $\alpha 1$ both for AIS and dendrites (Fig 6b). However, in Fig 5a, in neurons depleting $\beta 2$ spectrin, the $\alpha 1$ subunit signaling increases. Any explanations for that? Also, how about the $\alpha 2$ subunit blotting in neurons depleting $\beta 2$ spectrin?

8) It is interesting to find that reduction of $\beta 2$ spectrin increase the amplitude of inhibitory synaptic currents, but any potential mechanisms regarding this aspect are not discussed.

Minor points:

1) Keywords: keep the GABAA receptors uniform format;

2) Abstract: axon initial segment, there should be no "-" between "axon" and "initial", also be uniform in the following parts; interrogate should be "interrogated"; $\alpha 1$ should be " $\alpha 1$ " (the same on page 6);

3) Page5, the first paragraph last third row: "?2-containing..."

4) Fig S1 legend, "flow-through" instead of "flow-though";

5) GFP or Gfp, VGAT or Vgat, should be uniform throughout the manuscript.

6) The format (e.g. bold letter) for the figure legends are poorly prepared.

7) P8: the last sentence, should be "Fig 5b", instead of "Fig 4b".

The overall writing for this paper is insouciant. There are too many typos and errors to point out. The reviewer suggests a throughout revision of the writing that make the scientific findings elevated.

Reviewer #3 (Remarks to the Author):

The study by Smalley et al presents interesting results regarding the how subunit composition of GABAA receptors and their interaction with beta spectrin complexes influences their location (axon initial segment vs dendrites). In favor of the authors' conclusions, manipulation of a specific spectrin type (the beta 2 spectrin) significantly affects putative synapse numbers and the proportion of GABAA receptors containing the alpha 1 and alpha 2 subunits. The findings are interesting and novel and will likely be very influential in the field.

I have a few concerns regarding some aspects of experiments and data analysis.

Major:

1- there is no information about how the imaging analysis was performed. Colocalizations are shown by simply presenting a merged image, but no real statistical quantification is provided. Furthermore, no verification that the colocalization is non-random is provided. These issues apply to all figures reporting immunohistochemical data.

2- Figure 2. The puncta for VGAT and alpha 1 and alpha 2 are supposed to be opposing, but look completely overlapped. An analysis of the fluorescence across puncta would help assess whether peaks are indeed aligned or within the expected distance of pre and postsynaptic puncta.

3- Figure 7. The rationale for the multiGaussian fit in panel d is unclear. While the shift toward larger mIPSC amplitudes is pretty clear, its dependence on shifts in specific peaks of the distribution is questionable at best. The multi-peaks within the Sptbn1+/+ group appear to be part of a larger distribution that contains them all, and not representative of a real multi-peak distribution. The primary effect could simply be explained by a widening of the overall distribution with a shift in the median.

3- The number of cells included in the mIPSC analysis is very small and the average values/cell present quite a bit of variability, especially for the weighted decay and mIPSC frequency.

4- It is unclear how the electrophysiology data are interpreted against the molecular data. The rationale for this experimental component is not well explained. Are the authors assuming that the change in relative proportion of alpha 1 and alpha 2 subunits in the receptors drives the increase in amplitude of mIPSCs? Would that depend on the conductance of receptors with a different subunit composition, on the location of the receptors or on the interaction of the altered GABA receptors with intracellular protein complexes? Would it trigger compensatory changes in release probability?

Minor:

1- The ROI in Fig.4c appears misaligned with the dendritic process.

Dear Reviewers,

Thank you for your comprehensive review of our manuscript. Please find a point-by-point response to the issues raised. We look forward to your feedback.

Yours sincerely,
Stephen J. Moss

Reviewer 1 MAJOR COMMENTS

1. In Figure 5, the *sptbn1* ^{-/-} vs *sptbn1* ^{+/+} western blot was used to quantify loss of β 2 spectrin and an increase in β 4 spectrin. Yet the quantification of the other bands including GABAAR alpha 1 & alpha 2 subunits was not provided. This data should be provided, particularly as the GABAAR alpha 1 subunit appears increased in the representative blot data, and the alpha 2 subunit is unchanged. This is an important point to resolve as the IF data in Fig 6 shows increased alpha2 in AIS and dendrites and a 50% or greater loss of alpha 1 in dendrites and AIS. Is it possible alpha1 containing GABAARs are just not stabilized at synapses and accumulates intracellularly? The increase in alpha 2 clusters in dendrites & AIS should be confirmed by surface biotinylation or membrane fractions. This would help also in resolving why there is a 50% loss of inhibitory synapses in *sptbn1* ^{-/-} neurons (Fig 5b) at the same time as the rather surprising result that GABAergic inhibition is increased (mIPSC amplitude increase, Fig 7).

Reply

We apologize for the oversight. In our revised manuscript we have quantified the effects of SPTBN1 on GABA_AR subunit expression levels. As illustrated in Fig 5a of our revised manuscript, reducing SPTBN1 did not significantly modify the expression levels of the GABA_AR a1 or a2 subunits.

The mechanism by which spectrins regulate the synapse specific targeting of GABA_ARs could occur at multiple levels. For example, they could modify the endocytosis or exocytosis of individual receptor subtypes, in addition to regulating their assembly in the endoplasmic reticulum. Alternatively, spectrins may regulate the confinement of GABA_ARs at synapses. We have discussed these points in our revised manuscript.

We apologize for the lack of clarity; sIPSC are recorded with a somatic patch electrode. Using this approach, in non-spherical cells, such as neurons, the membrane potential is not clamped distal to the voltage-clamp electrode. Thus, currents recorded from a neuron are severely distorted due to the lack of space clamp, which in practice means that currents arising from the distal dendrites cannot be resolved using this technique (Spruston et al., 1993 J. Neurophysiology, 1993; 70: 781-802, Yehuda et al., 2008; J. Neurophysiology; 99(3):1127-36. Thus, sIPSCs measured using somatic patch clamp recording arise from the activation of synapses on the cell body and the AIS, but not those on dendrites. Therefore, the increase in sIPSC amplitude seen in Fig 7 are consistent with the increase in the number of inhibitory synapses containing the a2 subunit on the AIS of SPTBN1 deficient neurons (Fig 5).

These points have been discussed in our revised paper and the respective papers have been cited.

2. One would also predict that a large decrease in alpha 1 subunits should lead to a larger tau decay, however Fig 6 shows decay as unchanged. There should be a large reduction in

zolpidem sensitivity of the mIPSCs. This additional ephys experiment to verify loss of alpha 1 or alternatively a biochemical experiment showing enhanced surface levels of alpha 2 would strengthen and refine the conclusion. In general there is insufficient discussion of the ephys results combined with data from Fig 5 and Fig 6.

Reply

Early experiments based on $\alpha 1$ subunit knockout-mice reported that IPSC decay is slowed, but reciprocal experiments in $\alpha 2$ knockout mice have reported that their decay is unaltered (Goldstein et al.; J Neurophysiol, 2002 Dec;88(6):3208-17.). Thus, the relationship between increased/decreased subunit expression levels on sIPSC decay is complicated.

3. In its current form Figure 3 is not providing sufficient information. An excel supplementary table of the detected protein in the 3 different conditions should be provided, preferably with the assigned enriched GO terms that were used to generate Fig 3b. What was the criterion used for determining "significant enrichment" vs the IgG & 9e10 control background? "Only those proteins significantly enriched compared to control were included for downstream analysis. Using these criteria 121 proteins were detected co-purifying with the $\alpha 1$ subunit, while 123 were associated with $\alpha 2$, 45 of which were common to both purifications (Fig 3a)." Further description is needed in the methods, as none is currently provided.

Reply

We apologize for this oversight we have modified our manuscript and have providing details of how 'significant enrichment' was defined. The following has been added to the methods section: The Spectral Index normalized to the Global Index (SiGi) was calculated for each protein using all the detected peptides for each protein. A Welch t-test was performed comparing the SiGi for each protein detected by MS that immunoprecipitated with the $\alpha 1$ or $\alpha 2$ receptors compared to the SiGi in their equivalent control. For the $\alpha 1$ receptor IP, the control was non-immune IgG bound to protein G Dynabeads. For the $\alpha 2$ receptor IP, the control was anti-myc Dynabeads in wildtype tissue, rather than tissue from the pHa2 mouse. 'Significantly enriched' proteins were those with a subsequent p value lower than 0.05.

A table of the significantly enriched proteins has been included as supplementary information in our revised manuscript.

4. Insufficient statistical information: Several figure legends do not indicate n #s and this needs to be added. #s neurons from # cultures, and # experiments for IPs, etc. Except for Fig 4 that indicated n=4 for Fig 4b & representative images of 4 independent cultures (n=4) for 4c. Fig 5 only indicates n# for the western, but no info is provided for Fig 5b. Fig 7 provides sufficient info on n#.

Reply

We apologize for these oversights the respective information has been included in the legends for Figs 4-7.

MINOR COMMENTS

Results

It would be helpful to define pHa2 before you use it. Consider reordering/rearranging with the following sentence for clarity.

Reply

We have defined pHa2 and cited the appropriate publications; Nakamura et al., 2016 and 2020.

It would be helpful in evaluating the immunostaining if it was mentioned in the results that the neurons were permeabilized, so puncta are total dendritic alpha 1 & alpha 2, rather than surface clusters.

Reply

Neurons were permeabilized prior to immunostaining. This has been stated in the methods section and result sections of our revised manuscript.

Provide references for "This result agrees with previous studies suggesting a reciprocal relationship between the expression levels of these distinct spectrins."

Reply

We apologize for this oversight; the respective reference has been included in our revised manuscript: (Galiano, Jha et al. 2012) Figure 7 E and F.

It would be helpful to the general reader to indicate that the SPTBN1 gene encodes the β 2 spectrin protein.

Reply

We have stated that Spectrin b2 is encoded by the SPTBN1 gene.

Fig 4A: what is the 600 kDa complex with B4 spectrin in both lanes

Reply

This is likely to be a non-specific band, as it does not resolve at the same molecular weight as either GABA_AR subtype. It is not visible by Coomassie staining and is therefore of much lower abundance compared to those bands at 250kDa and 720kDa. We have mentioned this in the updated manuscript.

The following sentence needs to be revised: " α 2 spectrin co-purified both GABAAR subtypes while β 2 spectrin was enriched with those containing α 1 subunits." No α 2 spectrin IP was done or blotting in an IP reaction. Perhaps it was identified by mass spec, in which case the sentence should be reworded " α 2 spectrin was co-purified with both GABAAR subtypes" or alternatively this is a typo.

Reply

We apologize and this have been corrected to; "As measured by LC-MS/MS, α 2 spectrin co-purified both GABA_AR subtypes".

The authors should also mention the following reference: Cousin MA et al., Pathogenic SPTBN1 variants cause an autosomal dominant neurodevelopmental syndrome. Nature Genetics, 2021; DOI: 10.1038/s41588-021-00886-z

Reply

We have cited the respective publication in our revised manuscript.

TYPOS

We thank the referee and have corrected each typo detailed below.

Abstract:

Line 6 , a1 subunits

Line 10, "enriched GABAAR", replace with enriched with the $\alpha 2$ subunit

Line 15, "targeting of GABAAR," add s = GABAARs

Intro paragraph 2,

" $\alpha(1-6)$ $\beta(1-3)$," Insert , between (1-6) and β

"majority GABAARs" , insert of, majority of GABAARs

Results:

"functional", change to functionally

"Gfp", change to GFP (also in other areas of results section)

720kDa for $\beta 2$ -containing = presumably $\alpha 2$

"single a1 subunit isoforms (Fig 2c)." = replace a1 with $\alpha 1$

"Examining the role that $\beta 2$ spectrin plays" - make all heading text bold

"condition $\beta 2$ spectrin", change to conditional

" $\beta 2$ spectrin-/- neurons (Fig 4b)." = change to (Fig 5b)

"This was accompanied by a significant increase ($p < 0.05$) in the density of $\alpha 1$ puncta or approximately 50%", change to $\alpha 2$ puncta

"revealed modifications in clustering of GABAAR a subunit", change to α or alpha subunit

"Neurons have capacity" - change to Neurons have the capacity

Discussion:

"neurons have capacity", replace with neurons have the capacity

"to isolated native populations", replace with to isolate native....

"consisting of $\alpha 2$ spectrin and variable β spectrin that create" , replace with ... β spectrin, create

"Decreasing $\beta 2$ spectrin levels reduced the number of dendritic synapses containing the $\alpha 1$ subunit numbers containing GABAAR $\alpha 1$ subunits" , change to Decreasing $\beta 2$ spectrin levels reduced the number of dendritic synapses containing the GABAAR $\alpha 1$ subunits

"In contrast $\alpha 2$ subunit puncta on the AIS and dendrites assembled from $\alpha 2$ subunits.", I am not sure what the authors meant here.

Figure legends:

Fig 1 -There is a duplicate description for d)

Fig 4:"These were also probed for the presence of and $\beta 4$ spectrin to demonstrate that they are also present in complexes with $\alpha 1$ and $\alpha 2$ containing GABAARs." should be presence of and $\beta 2$ and $\beta 4$

Reviewer 2

Major points:

1) In the title, the authors claim that spectrin isoforms mediate the selective accumulation of GABAAR. However, this is not directly demonstrated. The data showed different spectrin isoforms co-purified with different α subunits, but not necessarily to be the reason for the selection. The reviewer suggests a more suitable title that represents the actual data.

Reply

We have amended the title to “Spectrin-beta 2 facilitates the selective accumulation of GABA_ARs at somatodendritic synapses”

2) Introduction: there are many conclusive descriptions of previous studies, but no reference papers were cited. For example, “Immunolocalization studies have revealed that principal neurons within the hippocampus express α 1-3, β 1-3 and γ 2 subunits...”, which immunolocalization studies and how they performed the experiments should be at least cited or described. It is the same situation for other paragraphs in the whole introduction part.

Reply

We apologize for this oversight citations and have been included in our revised manuscript.

3) Spectrins: the spectrin proteins are always considered as the cytoskeleton proteins that form tetrameric filaments providing the mechanical support. The authors have also cited the review paper (PMID:31191255) by Dr. Rasband, who is one of the authors for the present study. However, the authors considered spectrins as “scaffold” proteins instead of cytoskeleton proteins, which cannot be accepted by the reviewer. Furthermore, the Ankyrin family proteins, always forming duo complex with spectrins and functioning to linking membrane proteins to spectrin-based cytoskeleton, should be recognized as “scaffold” proteins, not simply described as the marker of the AIS. Although the authors made a short description in the “Discussion” part, the reviewer think it is worthy to make it clear.

Reply

Concordant with the reviewers' suggestions we have referred to spectrins as cytoskeletal proteins.

4) Fig. 1: page5, there is no descriptions or any references about how to make “knock-in mice in which pH sensitive GFP and the 9E10 epitope (pHa2) have been incorporate at the N-terminus of this subunit”, thus making it quite confusing and less convincing for the readers to appreciate these data. It would also help if a picture/panel could be present for the design of these experiments.

Reply

We have cited papers which describe the creation and characterization of the pHa2 mice.

Fig.1 panel b), d), and f), the statistics are confusing and not stated how to compare and calculate the p values as no data for the WT row is shown.

Reply

We have included a description of how the p values were calculated. In this instance the WT condition only refers to the purification performed to establish the background protein binding to the myc-beads for the calculation of significantly enriched proteins associated with pHa2, rather than a different experimental treatment condition. It is equivalent to the IgG condition for the a1 immunopurification.

Also, no LC-MS raw data is provided to justify the reliability of the results for Fig. 1.

Reply

The data in Figure 1 is accrued from multiple experiments for each receptor subtype. The respective spectral data will be deposited in the proteome X-change database prior to publication. We have added Table S1 for the binding proteins associated with the two receptor subtypes in higher molecular weight complexes.

For Fig. 1 legend, the last panel should be f), not d).

The d) panel, there is no “***” and “****” at all, so the description should be moved to f).

Reply

We apologize for this oversight, and we have corrected the labeling of this Figure.

5) Fig.4: panel a), what is the major band (between 480 and 720 kDa, lower than the 750 kDa band for β 4 spectrin) of the β 4 blot, α 2 line picture?

Reply

This is likely to be a non-specific band, as it does not resolve at the same molecular weight as either GABA_AR subtype. It is not visible by Coomassie staining and is therefore of much lower abundance compared to those bands at 250kDa and 720kDa. We have mentioned this in the updated manuscript.

Fig. 4 legend: a) I think there is a missing of β 2 in the fourth row; c) no scale bar length for the imaging on the right and no numbers are provided (also for Fig 2).

Reply

We have modified the figures and legends for figures 2 and 4.

6) Fig 5: panel a), it is quite strange for the statistical presentation for only one single column. Can the authors modify the comparison method to better present the results?

Reply

In this figure the data for SPTBN1^{-/-} has been normalized to SPTBN1^{+/+} cultures (100%), thus the single column format is appropriate.

7) The authors showed that reducing β 2 spectrin expression results in reducing the synaptic accumulation of the α 1 both for AIS and dendrites (Fig 6b). However, in Fig 5a, in neurons depleting β 2 spectrin, the α 1 subunit signaling increases. Any explanations for that? Also, how about the α 2 subunit blotting in neurons depleting β 2 spectrin?

Reply

We apologize for any confusion, newly added quantification of our immunoblots has revealed that the total expression levels of the α 1 subunit is not significantly modified in SPTBN1 deficient neurons (Fig 5a).

8) It is interesting to find that reduction of β 2 spectrin increase the amplitude of inhibitory synaptic currents, but any potential mechanisms regarding this aspect are not discussed.

Reply

In our revised manuscript we have discussed this issue in depth as detailed in “reply 1 to referee 1”.

Minor points:

Reply

We have corrected each typo and revised our manuscript accordingly.

- 1) Keywords: keep the GABAA receptors uniform format;
- 2) Abstract: axon initial segment, there should be no “-” between “axon” and “initial”, also be uniform in the following parts; interrogate should be “interrogated”; $\alpha 1$ should be “ $\alpha 1$ ” (the same on page 6);
- 3) Page 5, the first paragraph last third row: “ $\alpha 2$ -containing...”
- 4) Fig S1 legend, “flow-through” instead of “flow-though”;
- 5) GFP or Gfp, VGAT or Vgat, should be uniform throughout the manuscript.
- 6) The format (e.g. bold letter) for the figure legends are poorly prepared.
- 7) P8: the last sentence, should be “Fig 5b”, instead of “Fig 4b”.

The overall writing for this paper is insouciant. There are too many typos and errors to point out. The reviewer suggests a throughout revision of the writing that make the scientific findings elevated.

Reviewer 3

Major Points:

1- there is no information about how the imaging analysis was performed. Colocalizations are shown by simply presenting a merged image, but no real statistical quantification is provided. Furthermore, no verification that the colocalization is non-random is provided. These issues apply to all figures reporting immunohistochemical data.

Reply

We have provided further experimental details of how the immunolocalization experiments were performed.

2- Figure 2. The puncta for VGAT and $\alpha 1$ and $\alpha 2$ are supposed to be opposing, but look completely overlapped. An analysis of the fluorescence across puncta would help assess whether peaks are indeed aligned or within the expected distance of pre and postsynaptic puncta.

Reply

At the magnification shown it is not possible to distinguish between the pre- and postsynaptic compartments. VGAT staining in this Figure is simply used to differentiate synaptic GABA_ARs from their extra synaptic counterparts.

3- Figure 7. The rationale for the multiGaussian fit in panel d is unclear. While the shift toward larger mIPSC amplitudes is pretty clear, its dependence on shifts in specific peaks of the distribution is questionable at best. The multi-peaks within the Sptbn1^{+/+} group appear to be part of a larger distribution that contains them all, and not representative of a real multi-peak

distribution. The primary effect could simply be explained by a widening of the overall distribution with a shift in the median.

Reply

In our revised manuscript we have removed the Gaussian distributions in Figure 7.

4- The number of cells included in the mIPSC analysis is very small and the average values/cell present quite a bit of variability, especially for the weighted decay and mIPSC frequency.

Reply

The variability in these parameters has been noted. However, this does not affect the major finding of this experiment which reveals that mean sIPSC amplitude is significantly increased in SPTBN1 knock-out neurons.

5- It is unclear how the electrophysiology data are interpreted against the molecular data. The rationale for this experimental component is not well explained. Are the authors assuming that the change in relative proportion of alpha 1 and alpha 2 subunits in the receptors drives the increase in amplitude of mIPSCs? Would that depend on the conductance of receptors with a different subunit composition, on the location of the receptors or on the interaction of the altered GABA receptors with intracellular protein complexes? Would it trigger compensatory changes in release probability?

Reply

Please see our reply to the first question raised by reviewer 1. The increases in sIPSC amplitude reflect an increase in the number of GABA_ARs synapses, or an increase in their mean conductance.

Minor Points:

1- The ROI in Fig.4c appears misaligned with the dendritic process.

Reply

We have realigned the ROI in Fig 4c.

Replies to the referee's comments

We thank the referees for their insightful critiques. The modifications we have made to our manuscript in response to each of their comments are detailed below.

Reviewer 1

MAJOR COMMENTS

1. In Figure 5, the *sptbn1* ^{-/-} vs *sptbn1* ^{+/+} western blot was used to quantify loss of β 2 spectrin and an increase in β 4 spectrin. Yet the quantification of the other bands including GABAAR alpha 1 & alpha 2 subunits was not provided. This data should be provided, particularly as the GABAAR alpha 1 subunit appears increased in the representative blot data, and the alpha 2 subunit is unchanged. This is an important point to resolve as the IF data in Fig 6 shows increased alpha2 in AIS and dendrites and a 50% or greater loss of alpha 1 in dendrites and AIS. Is it possible alpha1 containing GABAARs are just not stabilized at synapses and accumulates intracellularly? The increase in alpha 2 clusters in dendrites & AIS should be confirmed by surface biotinylation or membrane fractions. This would help also in resolving why there is a 50% loss of inhibitory synapses in *sptbn1* ^{-/-} neurons (Fig 5b) at the same time as the rather surprising result that GABAergic inhibition is increased (mIPSC amplitude increase, Fig 7).

Reply

To address this issue, we have quantified the effects of Sptbn1 on GABA_AR subunit expression levels in WT and Sptbn1 knock-out neurons. After normalization to β -actin, the expression levels of the α 1 or α 2 subunits were comparable between genotypes, see; Fig 5a and lines 281-285 of our revised paper.

The mechanism by which spectrins regulate the synapse specific targeting of GABA_ARs could occur at multiple levels. For example, they could modify the endocytosis or exocytosis of individual receptor subtypes, in addition to regulating their assembly in the endoplasmic reticulum. Alternatively, spectrins may regulate the confinement of GABA_ARs at synapses. We have discussed these points in the discussion of our revised manuscript (see lines 371-383).

We apologize for the lack of clarity; sIPSC are recorded with a somatic patch electrode. Using this approach, in non-spherical cells, such as neurons, the membrane potential is not clamped distal to the voltage-clamp electrode. Thus, currents recorded from a neuron are severely distorted due to the lack of space clamp, which in practice means that currents arising from the distal dendrites cannot be resolved using this technique (Spruston et al., 1993; J. Neurophysiology, 1993; 70: 781-802, Yehuda et al., 2008; J. Neurophysiology; 99(3):1127-36. Thus, sIPSCs measured using somatic patch clamp recording arise from the activation of synapses on the cell body and the AIS, but not those on dendrites. Therefore, the increase in sIPSC amplitude seen in Fig 7 are consistent with the increase in the number of inhibitory synapses containing the α 2 subunit on the AIS of Sptbn1 deficient neurons (Fig 5).

These points have been discussed in our revised paper and the respective papers have been cited.

2. One would also predict that a large decrease in alpha 1 subunits should lead to a larger tau decay, however Fig 6 shows decay as unchanged. There should be a large reduction in zolpidem sensitivity of the mIPSCs. This additional ephys experiment to verify loss of alpha 1 or

alternatively a biochemical experiment showing enhanced surface levels of alpha 2 would strengthen and refine the conclusion. In general, there is insufficient discussion of the ephys results combined with data from Fig 5 and Fig 6.

Reply

In our revised manuscript we demonstrate that the expression levels of the $\alpha 1$ and $\alpha 2$ subunits are not modified in *Sptbn2* knockout neurons (Fig 5). The relationship between the $\alpha 1$ subunit and mIPSC decay is complex. Early, experiments based on $\alpha 1$ subunit knockout-mice reported that IPSC decay is slowed, but reciprocal experiments in $\alpha 2$ knockout mice have reported that their decay is unaltered (Goldstein et al.; J Neurophysiol, 2002 Dec;88(6):3208-17.).

3. In its current form Figure 3 is not providing sufficient information. An excel supplementary table of the detected protein in the 3 different conditions should be provided, preferably with the assigned enriched GO terms that were used to generate Fig 3b. What was the criterion used for determining "significant enrichment" vs the IgG & 9e10 control background? "Only those proteins significantly enriched compared to control were included for downstream analysis. Using these criteria 121 proteins were detected co-purifying with the $\alpha 1$ subunit, while 123 were associated with $\alpha 2$, 45 of which were common to both purifications (Fig 3a)." Further description is needed in the methods, as none is currently provided.

Reply

We apologize for this oversight we have modified our manuscript and have providing details of how 'significant enrichment' was defined. The following has been added to the methods section: The Spectral Index normalized to the Global Index (SiGi) was calculated for each protein using all the detected peptides for each protein. A Welch t-test was performed comparing the SiGi for each protein detected by MS that immunoprecipitated with the $\alpha 1$ or $\alpha 2$ receptors compared to the SiGi in their equivalent control. For the $\alpha 1$ receptor IP, the control was non-immune IgG bound to protein G Dynabeads. In the case of the $\alpha 2$ purifications the control was anti-myc Dynabeads in wildtype tissue, rather than tissue from the pHa2 mouse. 'Significantly enriched' proteins were those with a subsequent p value lower than 0.05.

A table of the significantly enriched proteins has been included as supplementary information in our revised manuscript.

4. Insufficient statistical information: Several figure legends do not indicate n #s and this needs to be added. #s neurons from # cultures, and # experiments for IPs, etc. Except for Fig 4 that indicated n=4 for Fig 4b & representative images of 4 independent cultures (n=4) for 4c. Fig 5 only indicates n# for the western, but no info is provided for Fig 5b. Fig 7 provides sufficient info on n#.

Reply

We apologize for these oversights the respective information has been included in the legends for Figs 4-7.

MINOR COMMENTS

Results

It would be helpful to define pHa2 before you use it. Consider reordering/rearranging with the following sentence for clarity.

Reply

We have defined pHa2 and cited the appropriate publications; (Nakamura et al., 2016 and 2020.

It would be helpful in evaluating the immunostaining if it was mentioned in the results that the neurons were permeabilized, so puncta are total dendritic alpha 1 & alpha 2, rather than surface clusters.

Reply

Neurons were permeabilized prior to immunostaining. This has been stated in the methods section and result sections of our revised manuscript.

Provide references for "This result agrees with previous studies suggesting a reciprocal relationship between the expression levels of these distinct spectrins."

Reply

We apologize for this oversight; the respective reference has been included in our revised manuscript: (Galiano, Jha et al. 2012) Figure 7 E and F.

It would be helpful to the general reader to indicate that the SPTBN1 gene encodes the $\beta 2$ spectrin protein.

Reply

We have stated that Spectrin b2 is encoded by the SPTBN1 gene.

Fig 4A: what is the 600 kDa complex with B4 spectrin in both lanes

Reply

This is likely to be a non-specific band, as it does not resolve at the same molecular weight as either GABA_AR subtype. It is not visible by Coomassie staining and is therefore of much lower abundance compared to those bands at 250kDa and 720kDa. We have mentioned this in the updated manuscript.

The following sentence needs to be revised: " $\alpha 2$ spectrin co-purified both GABAAR subtypes while $\beta 2$ spectrin was enriched with those containing $\alpha 1$ subunits." No $\alpha 2$ spectrin IP was done or blotting in an IP reaction. Perhaps it was identified by mass spec, in which case the sentence should be reworded " $\alpha 2$ spectrin was co-purified with both GABAAR subtypes" or alternatively this is a typo.

Reply

We apologize and this have been corrected to; "As measured by LC-MS/MS, $\alpha 2$ spectrin co-purified both GABA_AR subtypes".

The authors should also mention the following reference: Cousin MA et al., Pathogenic SPTBN1 variants cause an autosomal dominant neurodevelopmental syndrome. Nature Genetics, 2021; DOI: 10.1038/s41588-021-00886-z

Reply

We have cited the respective publication in our revised manuscript.

TYPOS

We thank the referee and have corrected each typo detailed below.

Abstract:

Line 6 , $\alpha 1$ subunits

Line 10, "enriched GABAAR", replace with enriched with the $\alpha 2$ subunit

Line 15, "targeting of GABAAR," add s = GABAARs

Intro paragraph 2,

" $\alpha(1-6) \beta(1-3)$," Insert , between (1-6) and β

"majority GABAARs" , insert of, majority of GABAARs

Results:

"functional", change to functionally

"Gfp", change to GFP (also in other areas of results section)

720kDa for $\beta 2$ -containing = presumably $\alpha 2$

"single $\alpha 1$ subunit isoforms (Fig 2c)." = replace $\alpha 1$ with $\alpha 1$

"Examining the role that $\beta 2$ spectrin plays" - make all heading text bold

"condition $\beta 2$ spectrin", change to conditional

" $\beta 2$ spectrin-/- neurons (Fig 4b)." = change to (Fig 5b)

"This was accompanied by a significant increase ($p < 0.05$) in the density of $\alpha 1$ puncta or approximately 50%", change to $\alpha 2$ puncta

"revealed modifications in clustering of GABAAR α subunit", change to α or alpha subunit

"Neurons have capacity" - change to Neurons have the capacity

Discussion:

"neurons have capacity", replace with neurons have the capacity

"to isolated native populations", replace with to isolate native....

"consisting of $\alpha 2$ spectrin and variable β spectrin that create" , replace with ... β spectrin, create

"Decreasing $\beta 2$ spectrin levels reduced the number of dendritic synapses containing the $\alpha 1$ subunit numbers containing GABAAR $\alpha 1$ subunits" , change to Decreasing $\beta 2$ spectrin levels reduced the number of dendritic synapses containing the GABAAR $\alpha 1$ subunits

"In contrast $\alpha 2$ subunit puncta on the AIS and dendrites assembled from $\alpha 2$ subunits.", I am not sure what the authors meant here.

Figure legends:

Fig 1 -There is a duplicate description for d)

Fig 4:"These were also probed for the presence of and $\beta 4$ spectrin to demonstrate that they are also present in complexes with $\alpha 1$ and $\alpha 2$ containing GABAARs." should be presence of and $\beta 2$ and $\beta 4$

Reviewer 2

Major points:

1) In the title, the authors claim that spectrin isoforms mediate the selective accumulation of GABAAR. However, this is not directly demonstrated. The data showed different spectrin isoforms co-purified with different α subunits, but not necessarily to be the reason for the selection. The reviewer suggests a more suitable title that represents the actual data.

Reply

We have amended the title to “Spectrin-beta 2 facilitates the selective accumulation of GABA_ARs at somatodendritic synapses”

2) Introduction: there are many conclusive descriptions of previous studies, but no reference papers were cited. For example, “Immunolocalization studies have revealed that principal neurons within the hippocampus express α 1-3, β 1-3 and γ 2 subunits...”, which immunolocalization studies and how they performed the experiments should be at least cited or described. It is the same situation for other paragraphs in the whole introduction part.

Reply

We apologize for this oversight citations and have been included in our revised manuscript.

3) Spectrins: the spectrin proteins are always considered as the cytoskeleton proteins that form tetrameric filaments providing the mechanical support. The authors have also cited the review paper (PMID:31191255) by Dr. Rasband, who is one of the authors for the present study. However, the authors considered spectrins as “scaffold” proteins instead of cytoskeleton proteins, which cannot be accepted by the reviewer. Furthermore, the Ankyrin family proteins, always forming duo complex with spectrins and functioning to linking membrane proteins to spectrin-based cytoskeleton, should be recognized as “scaffold” proteins, not simply described as the marker of the AIS. Although the authors made a short description in the “Discussion” part, the reviewer think it is worthy to make it clear.

Reply

Concordant with the reviewers' suggestions we have referred to spectrins as cytoskeletal proteins.

4) Fig. 1: page5, there is no descriptions or any references about how to make “knock-in mice in which pH sensitive GFP and the 9E10 epitope (pHa2) have been incorporate at the N-terminus of this subunit”, thus making it quite confusing and less convincing for the readers to appreciate these data. It would also help if a picture/panel could be present for the design of these experiments.

Reply

We have cited papers which describe the creation and characterization of the pHa2 mice.

Fig.1 panel b), d), and f), the statistics are confusing and not stated how to compare and calculate the p values as no data for the WT row is shown.

Reply

We have included a description of how the p values were calculated. In this instance the WT condition only refers to the purification performed to establish the background protein binding to the myc-beads for the calculation of significantly enriched proteins associated with pHa2, rather than a different experimental treatment condition. It is equivalent to the IgG condition for the a1 immunopurification.

Also, no LC-MS raw data is provided to justify the reliability of the results for Fig. 1.

Reply

The data in Figure 1 is accrued from multiple experiments for each receptor subtype. The respective spectral data will be deposited in the proteome X-change database prior to

publication. We have added Table S1 for the binding proteins associated with the two receptor subtypes in higher molecular weight complexes.

For Fig. 1 legend, the last panel should be f), not d).

The d) panel, there is no “**” and “***” at all, so the description should be moved to f).

Reply

We apologize for this oversight, and we have corrected the labeling of this Figure.

5) Fig.4: panel a), what is the major band (between 480 and 720 kDa, lower than the 750 kDa band for β 4 spectrin) of the β 4 blot, α 2 line picture?

Reply

This is likely to be a non-specific band, as it does not resolve at the same molecular weight as either GABA_AR subtype. It is not visible by Coomassie staining and is therefore of much lower abundance compared to those bands at 250kDa and 720kDa. We have mentioned this in the updated manuscript.

Fig. 4 legend: a) I think there is a missing of β 2 in the fourth row; c) no scale bar length for the imaging on the right and no numbers are provided (also for Fig 2).

Reply

We have modified the figures and legends for figures 2 and 4.

6) Fig 5: panel a), it is quite strange for the statistical presentation for only one single column. Can the authors modify the comparison method to better present the results?

Reply

In this figure the data for SPTBN1^{-/-} has been normalized to SPTBN1^{+/+} cultures (100%), thus the single column format is appropriate.

7) The authors showed that reducing β 2 spectrin expression results in reducing the synaptic accumulation of the α 1 both for AIS and dendrites (Fig 6b). However, in Fig 5a, in neurons depleting β 2 spectrin, the α 1 subunit signaling increases. Any explanations for that? Also, how about the α 2 subunit blotting in neurons depleting β 2 spectrin?

Reply

We apologize for any confusion, newly added quantification of our immunoblots has revealed that the total expression levels of the α 1 subunit is not significantly modified in SPTBN1 deficient neurons (Fig 5a).

8) It is interesting to find that reduction of β 2 spectrin increase the amplitude of inhibitory synaptic currents, but any potential mechanisms regarding this aspect are not discussed.

Reply

In our revised manuscript we have discussed this issue in depth as detailed in “reply 1 to referee 1”.

Minor points:

Reply

We have corrected each typo and revised our manuscript accordingly.

- 1) Keywords: keep the GABAA receptors uniform format;
- 2) Abstract: axon initial segment, there should be no "-" between "axon" and "initial", also be uniform in the following parts; interrogate should be "interrogated"; a1 should be " $\alpha 1$ " (the same on page 6);
- 3) Page5, the first paragraph last third row: "?2-containing..."
- 4) Fig S1 legend, "flow-through" instead of "flow-thought";
- 5) GFP or Gfp, VGAT or Vgat, should be uniform throughout the manuscript.
- 6) The format (e.g. bold letter) for the figure legends are poorly prepared.
- 7) P8: the last sentence, should be "Fig 5b", instead of "Fig 4b".

The overall writing for this paper is insouciant. There are too many typos and errors to point out. The reviewer suggests a throughout revision of the writing that make the scientific findings elevated.

Reviewer 3

Major Points:

1- there is no information about how the imaging analysis was performed. Colocalizations are shown by simply presenting a merged image, but no real statistical quantification is provided. Furthermore, no verification that the colocalization is non-random is provided. These issues apply to all figures reporting immunohistochemical data.

Reply

Subunit co-localization was quantified using the synapse counter plugin in ImageJ; see lines 163-164 of our manuscript as detailed previously (Kontou, Antonoudiou et al. 2021) (Dzyubenko, Rozenberg et al. 2016). In the methods section our revised paper, the analysis procedure used has been described (see lines 506-513).

2- Figure 2. The puncta for VGAT and alpha 1 and alpha 2 are supposed to be opposing, but look completely overlapped. An analysis of the fluorescence across puncta would help assess whether peaks are indeed aligned or within the expected distance of pre and postsynaptic puncta.

Reply

At the magnification shown it is not possible to distinguish between the pre- and postsynaptic compartments. VGAT staining in this Figure is simply used to differentiate synaptic GABA_ARs from their extra synaptic counterparts.

3- Figure 7. The rationale for the multiGaussian fit in panel d is unclear. While the shift toward larger mIPSC amplitudes is pretty clear, its dependence on shifts in specific peaks of the distribution is questionable at best. The multi-peaks within the Sptbn1^{+/+} group appear to be

part of a larger distribution that contains them all, and not representative of a real multi-peak distribution. The primary effect could simply be explained by a widening of the overall distribution with a shift in the median.

Reply

In our revised manuscript we have removed the Gaussian distributions in Figure 7.

4- The number of cells included in the mIPSC analysis is very small and the average values/cell present quite a bit of variability, especially for the weighted decay and mIPSC frequency.

Reply

The experimental numbers chosen were determined using power analysis based on previous studies that we have performed on cultured neurons (Nakamura et al., 2016, Nathanson et al., 2019 Kontou et al., 2021). This has been detailed in the methods of our revised manuscript (see lines 533-536).

5- It is unclear how the electrophysiology data are interpreted against the molecular data. The rationale for this experimental component is not well explained. Are the authors assuming that the change in relative proportion of alpha 1 and alpha 2 subunits in the receptors drives the increase in amplitude of mIPSCs? Would that depend on the conductance of receptors with a different subunit composition, on the location of the receptors or on the interaction of the altered GABA receptors with intracellular protein complexes? Would it trigger compensatory changes in release probability?

Reply

Please see our reply to the first question raised by reviewer 1. The increases in sIPSC amplitude reflect an increase in the number of GABA_ARs synapses, or an increase in their mean conductance.

Minor Points:

1- The ROI in Fig.4c appears misaligned with the dendritic process.

Reply

We have realigned the ROI in Fig 4c.

REVIEWERS' COMMENTS:

Reviewer #1 (Remarks to the Author):

In this insightful manuscript "Spectrin isoforms mediate the selective accumulation of GABA_A receptors at somatodendritic and axo-axonic synapses " the authors contribute multiple important findings related to GABA_ARs, the main receptor for generating fast inhibition in the CNS. Due to the diverse functions and large number of subunits that can form GABA_AR heteropentamers, defining the mechanisms of subtype sub-cellular compartmentalization is an important area of research and potential therapeutic drug development. Proteomics was used here to reveal that the majority of α 2-containing receptors co-assemble with α 1 subunits, whereas α 1 subunits can form GABA_ARs as the sole α subunit. In addition, proteomic and bioinformatic analysis identified the unique interactomes for α 1, α 2, and the overlapping α 1 α 2 GABA_AR interactome. Building on the specific interactome IDs, biochemical and immunofluorescence approaches were used to delineate GABA_AR interactions with distinct isoforms of spectrin, the neuronal scaffolds that link transmembrane proteins to the cytoskeleton. β 2 spectrin containing complexes were shown to preferentially associate with α 1 GABA_ARs, while β 4 spectrin associated with α 2 GABA_AR subtypes enriched on the AIS (axon initial segment). These associations occurred via the respective intracellular regions of the α 1 and α 2 subunits. Next, AAV mediated knockdown of β 2 spectrin resulted in a decrease in pre and post synaptic components of inhibitory synapses, and a loss of α 1 puncta in the AIS and dendrites. Interestingly, β 4 spectrin total protein levels were increased, concurrent with an upregulation of α 2 puncta in the in AIS and dendrites. Intriguingly the net change of these results led to enhanced GABAergic inhibition as seen from an increase in sIPSC amplitude. The observed perturbations of inhibitory synapses with knockdown of β 2 spectrin suggest potential GABA_AR contributions to the debilitating neurological phenotypes of spectrinopathies and recently identified neurodevelopmental syndromes in pathogenic SPTBN1 patient variants.

This is a valuable and interesting manuscript that should be impactful both to the neuroscience community and other fields (nervous system disorders/neuropharmacology, cytoskeletal/structural fields). Furthermore knowledge of spectrin function in dendrites is limited and identifying key interactors is critical to understanding general neuronal architecture and functional plasticity.

The authors have addressed most of my main concerns. Some very minor editorial revisions still need to be made.

1) Entire MS including methods, results, discussion & figure legends only describes mIPSCs - there is no mention of sIPSC.

Although the author indicated sIPSC in the rebuttal.

We apologize for the lack of clarity; sIPSC are recorded with a somatic patch electrode. Using this approach, in non-spherical cells, such as neurons, the membrane potential is not clamped distal to the voltage-clamp electrode.....

2) Figure 3 in the resubmitted MS overall is now of insufficient resolution and cannot be read, 3b needs to be larger (can be arranged vertically), 3a can be shrunk in size.

3) In Figure 7b) there are multiple-colored dots for the sptbn1 -/- data points instead of all being pale purple.

Reviewer #2 (Remarks to the Author):

The revised manuscripts address my original concerns and now I support the publication. Congrats on the beautiful study!

Reviewer #3 (Remarks to the Author):

The authors have thoroughly addressed my comments. I do not have additional concerns.

George Inglis, PhD
Senior Editor
Communications Biology
Senior Editor

Re COMMSBIO-22-0737B-Z.

Many thanks for the email regarding the above manuscript please find our responses to the issues raised by Referee 1 which are detailed below.

1. Entire MS including methods, results, discussion & figure legends only describes mIPSCs - there is no mention of sIPSC.

Although the author indicated sIPSC in the rebuttal.

We apologize for the lack of clarity; sIPSC are recorded with a somatic patch electrode. Using this approach, in non-spherical cells, such as neurons, the membrane potential is not clamped distal to the voltage-clamp electrode.....

We apologize, we measured miniature inhibitory synaptic currents and the term sIPSC has been replaced throughout the text with mIPSC.

2. Figure 3 in the resubmitted MS overall is now of insufficient resolution and cannot be read, 3b needs to be larger (can be arranged vertically), 3a can be shrunk in size.

We have enlarged and revised this figure to resolve the issues raised.

3) In Figure 7b) there are multiple-colored dots for the sptbn1 -/- data points instead of all being pale purple.

As requested, data points for sptbn1 -/- are now shown only in pale purple.

We hope that with these minor modifications are paper is now acceptable for publication.

Yours sincerely

Stephen J Moss Ph.D.